# Distilling noise characteristics and prior expectations in multisensory causal inference

**Shuze Liu**[1⊙], **Trevor Holland**[2⊙], **Wei Ji Ma**[2,3‡], **Luigi Acerbi**[4‡*]

**1** PhD Program in Neuroscience, Harvard University, Cambridge, Massachusetts, United States of America, **2** Previously at Department of Neuroscience, Baylor College of Medicine, Houston, Texas, United States of America, **3** Center for Neural Science and Department of Psychology, New York University, New York City, New York, United States of America, **4** Department of Computer Science, University of Helsinki, Helsinki, Uusimaa, Finland

⊙ These authors are primary authorship and equally contribution to this work.
‡ These authors are senior authorship and equally contribution to this work.
* luigi.acerbi@helsinki.fi

## Abstract

The perception of the external world relies on integrating information from multiple sensory modalities. To do this effectively, the brain must determine whether sensory signals come from a common source and, if so, combine them to reduce perceptual uncertainty. While Bayesian observer models have been successful in accounting for multisensory causal inference decisions by humans, they typically rely on simplifying assumptions that may not reflect the true complexity of human perception. In this study, we challenge two assumptions common in Bayesian multisensory perception models: homoskedastic (constant across space) sensory noise and Gaussian priors. We collected an auditory-visual perceptual dataset featuring both unisensory and bisensory tasks, where participants must either provide stimulus location estimates or same-different source judgments. Subsequently, we developed a flexible semi-parametric approach that allowed us to infer the sensory noise and prior shapes from participants' data, and subsequently 'distill' them into new model classes through visual inspection of the semiparametrically fitted function shapes. We find that human multisensory perception is best described by an eccentricity-dependent sensory noise that plateaus in the periphery and a prior distribution with a narrow central peak and smoother tails. We also found evidence for auditory range recalibration and increased sensory noise in multisensory conditions, suggesting complex interactions between sensory modalities. These findings deviate substantially from traditional modeling assumptions and highlight the value of data-driven rather than theory-driven modeling assumptions. Overall, our study demonstrates the value of systematically exploring model assumptions in multisensory research and provides a new set of modeling tools for perceptual causal inference.

**Data availability statement:** All data and analysis code are available at https://github.com/LSZ2001/Audiovisual-causal-inference.

**Funding:** LA was partly supported by the Research Council of Finland (grants 356498 and 358980). The funders had no role in study design, data collection and analysis, decision to publish, or preparation of the manuscript.

**Competing interests:** The authors have declared that no competing interests exist.

## Author summary

While interacting with the world, organisms are constantly exposed to stimuli from multiple sensory modalities. They would benefit from knowing whether multisensory stimuli share the same source, and, if so, from combining them. While Bayesian observer models have been developed to account for such cognitive processes, past literature has not fully addressed the formulation of two crucial model components—sensory noise and prior—through empirically-based approaches. We have developed more flexible Bayesian models that allow comprehensive exploration of sensory noise and prior shapes, using a set of human multisensory localization and causal inference tasks. We have then developed Bayesian models that are similarly constrained as models in previous works, but are instead inspired by the sensory noise and prior shapes found via our more flexible models. These constrained models offer a better description of human behavior compared to conventional Bayesian models based on the field's common assumptions on sensory noise and prior shapes. Our results reveal the complexity of human multisensory perception beyond past modeling assumptions, and highlight the efficacy of our new method—starting with more flexible Bayesian models—in objectively assessing these assumptions.

## Introduction

### Background

As we perceive and interact with the world around us, our brain is constantly gathering and processing information from multiple sensory modalities to make inferences about our external environment. For example, imagine yourself as a pre-modern hunter pursuing birds hiding in a tree. You may see movements of a branch indicative of a bird's presence, yet the visual scenery alone could not provide precise spatial information to decide where to shoot your arrow. Around the same time, you may also hear a chirp coming from the tree, likely coming from the same bird that is shaking the branch. Combining both pieces of information may give you a better idea of the specific location of the bird, and lead to greater chances of securing prey. Due to uncertainty inherent in sensory perception, the combination of multiple pieces of sensory information (cues) from a common stimulus would afford greater precision and more accurate estimates of that stimulus, compared to processing each piece of information alone [1–10]. In short, properly combining cues could make the difference between procuring dinner or an empty stomach.

However, in the hunter story above, the movement of the branch could be due to wind—a distant source independent of the hidden bird. If this were the case, it would be detrimental for the hunter to factor branch movements into bird localization. This example illustrates the importance of causal inference—judging whether distinct pieces of observations (e.g., visual and auditory) are generated by the same latent cause. Skillful manipulations can induce observers to erroneously believe that distinct multisensory stimuli share a single, common cause, such as in the ventriloquist illusion [4,11] and McGurk effect [12].

These illusions highlight the role of causal inference in assessing the same-source assumption [1,10,13–18], as the result of such inference determines whether sensory cues from different modalities should be integrated or kept separate during subsequent inference (e.g., inferring the location of one cue, or making sense of the sound being heard).

While the study of multisensory perception in humans dates back more than a century [19–22], the study of causal inference during multisensory perception is more recent [1]. Previous work has formalized both causal inference and multisensory perception in the language of Bayesian probabilistic inference [10,23], a framework proven to be highly successful in describing human perception across a variety of contexts including multisensory spatial localization [1,2,4,14,18,24–26], orientation perception [27], depth perception [28], as well as heading perception [29,30]. Specifically in the domain of multisensory spatial localization, cue integration breaks down when there is significant spatial disparity between cues [1,28]. In contrast, when cue disparity is small, humans weigh their noisy measurements of each cue according to their reliability in reaching a final location estimate [2,5,6]. These findings align with the *Bayesian causal inference* framework—when two stimuli are very far apart, they would likely generate noisy measurements that are also far apart, which would allow humans to infer that the underlying cues do not share the same location and should not be integrated; vice versa, two stimuli that are close to each other are more likely to generate similar measurements, leading to integration [1,10,13,14,31]. The success of Bayesian observer models in capturing human multisensory perception has motivated parallel neuroscience studies in animals [32,33], the development of neural computational models [34–36] as well as neuroimaging studies [37–39].

## Prior and noise assumptions

Bayesian observer models are built out of three pieces of information: priors, likelihoods, and measurement noise distributions [10]. In multisensory spatial localization tasks, the prior distribution over stimulus locations and the prior probability that two observations share the same source reflect the observer's statistical knowledge of the stimuli, before evaluating the noisy observations. The measurement noise distribution encodes how sensory observations are generated by the true stimuli, and represents an objective property of the system, such as how perceptual sensitivity depends on stimulus features (e.g., modality, contrast, location). Conversely, the likelihood function represents what the observer believes about the generative process, which describes how the true world structure (stimulus location, whether two observations indeed share the same stimulus source) generates observations corrupted by the observer's sensory noise [10]. In practice, it is generally assumed that the observer's sensory likelihood matches the sensory noise model, with occasional studies investigating the potential mismatch [40], so the Bayesian observer modeling problem reduces to defining two key elements: the prior and the noise model.

Given this formalism, the success of Bayesian multisensory observer models rests upon the adoption by researchers of priors and sensory noise structures that reflect the actual performance of sensory systems as well as the expectations held by human observers. Past studies often assume a Gaussian [1,10,25,26,41,42], uniform [2,31,43], or Gaussian-uniform mixture [14,44] prior over stimulus locations, coupled with constant ("homoskedastic") sensory noise across space [1,2,10,14,17,18,25,26,28,41–45]. These modeling assumptions conveniently lead to analytically tractable Bayesian inference [1,10,46], but may not adequately capture the complexities of human multisensory perception.

For example, previous works in heading direction and spatial localization have revealed that visual and vestibular estimation variability and discrimination thresholds vary with stimulus location [47–49], being lowest in the straight-ahead direction and higher in the periphery [10,13,25,41]. In other words, if we formulate the noisy observation as a random variable $x$ centered at the true stimulus location $s$, the random variable's variance $\sigma^2$ is dependent on its mean, a property known as unequal-variance or "heteroskedasticity" [10,50,51]. Such heteroskedastic noise can be expressed via a sensory noise function $\sigma(s)$ which specifies the standard deviation of the noise as a function of stimulus location $s$.

The few modeling works that incorporate heteroskedastic sensory noise often do so by assuming a fixed (parametric) shape, such as a quadratic family for $\sigma(s)$ [13], or by decomposing $\sigma(s)$ into two components, one being the baseline and the other component varying periodically as a function of stimulus angle [29,30]. As a result, the broader space of possible

prior or sensory noise profiles has not been systematically examined for their ability to account for human behavior. Recent work has developed flexible approaches to inferring priors and sensory noise from behavior, particularly within efficient coding frameworks that link these components normatively to environmental statistics [52–55]. However, these studies have focused primarily on unisensory estimation tasks. In the multisensory setting, priors and sensory noise interact with an additional layer of inference—the causal structure linking sensory signals—which introduces new complexities not addressed by existing frameworks. Our work extends flexible, data-driven inference of priors and noise to this richer setting.

In sum, while existing work has made important progress with standard modeling assumptions about priors and noise, the space of possible shapes has not been systematically explored in multisensory perception, motivating a more data-driven approach.

## Contributions and outline

Our study addresses the gap in past literature by exploring the prior and sensory noise function families that human observers use in multisensory perception in a comprehensive and systematic manner. Our approach is similar in spirit to previous work in other domains of Bayesian perception, such as visual speed perception [56], visual orientation perception [57], time perception [58], and spatial estimation [40], among others, which aimed at inferring the priors and noise structure directly from the data, with relatively little assumptions about their shapes. To our knowledge, ours is the first study attempting such reconstruction in multisensory causal inference.

To this end, we conducted an auditory-visual perception experiment containing a rich variety of perceptual judgments, including both unisensory and bisensory stimulus localization tasks as well as a bisensory causal inference task. The inclusion of unisensory tasks will allow us to refine our search for priors and sensory noise structures within each modality, before applying them to bisensory causal inference and cue integration. While some previous spatial localization studies use left-right reports [13] or a discrete set of possible responses [1,2], our localization tasks require the placement of a cursor on a near-continuous range [25], providing fine-grained responses.

Our modeling goals are twofold:

1. First, we aim at extracting from the data the sensory noise functions and prior distributions that adequately describe human causal inference and stimulus localization. We achieve this through the usage of flexible *semiparametric* Bayesian observer models. These models represent prior probability density functions (pdfs) and sensory noise functions via their values defined at pre-established "pivot" points within the stimulus range (hence "semiparametric"), followed by smooth interpolation [56,58]. In short, these semiparametric models can freely approximate any smooth curve and would thus capture the shape of prior and noise directly from the data.

2. Our second aim is to derive a model family for priors and noise that captures the complexity of multisensory perception, without the need to go through the complexity of the full semiparametric approach. To this aim, we "distill" a new *parametric* model family inspired by our semiparametric results. We empirically validate the superiority of such distilled parametric models, compared to conventional parametric models, in explaining human multisensory perception.

In the following sections, we will first introduce our experiment design, featuring both unimodal and bimodal spatial localization as well as bimodal causal inference tasks. Second, we will present results from our semiparametric model fits of unisensory data, which confirm the location-dependence of visual and auditory sensory noise as well as a prior that features a 'central peak', favoring straight-ahead responses in the perceptual estimation task as visible in the data. Third, we will show that the semiparametrically fitted sensory noise and prior shapes generalize well to bisensory causal inference and spatial localization tasks. Finally, we will 'distill' parametric model families from the semiparametric sensory noise and prior shapes, and fit these distilled models to both unisensory and bisensory data.

To provide a faithful account of human causal inference and bisensory perception, we adjudicate among candidate models through quantitative model comparison. The inclusion of additional model parameters (allowing for heteroskedastic sensory noise and non-Gaussian priors) would inevitably capture human behavior better than the baseline Bayesian model nested within. However, an unlimited expansion of model components is also undesirable, as it sacrifices interpretability and increases the risk of overfitting [59]. Following common practice in computational modeling, we utilize the Akaike and Bayesian Information Criteria (AIC/BIC) [60]. Both these model comparison methods penalize the number of free parameters on top of each model's goodness-of-fit, thereby striking a balance between simplicity and explanatory power [61]. That said, statistical model comparison alone does not fully address concerns about flexibility: a model may be statistically favored yet remain too unconstrained to constitute a meaningful theory. We therefore adopt two complementary strategies beyond AIC/BIC: first, we estimate key model components from unisensory data and "lift" them to bisensory conditions—requiring the model to generalize these components across task contexts rather than fitting each independently; second, we "distill" parsimonious parametric model families from the shapes revealed by flexible semiparametric fits.

Our model comparison results indicate that an exponentially shaped sensory noise function that dips in the center stimulus range and plateaus on the peripheries, coupled with a prior that combines a narrow peak with a broader smooth distribution (technically, a Gaussian-Laplace mixture prior), best describe our human data. These findings demonstrate the complexity of human causal inference and stimulus localization and highlight the utility of data-oriented semiparametric modeling approaches for reaching realistic modeling assumptions.

## Results

### Experiment

To systematically investigate the shape of sensory noise and priors used by human participants during multisensory causal inference and stimulus localization, we designed an experiment that would test our participants' perceptual strategies in a rich variety of tasks. The experiment followed the Declaration of Helsinki and was approved by the Institutional Review Board of Baylor College of Medicine.

Our experiment consisted of five tasks done by each participant (Fig 1A). In the *unisensory visual localization* (UV) and *unisensory auditory localization* (UA) tasks, participants were asked to estimate the location of a single visual or auditory stimulus, respectively. In the *bisensory causal judgment* (BC) task, participants were simultaneously presented with one visual stimulus and one auditory stimulus, and were asked to report whether the two stimuli shared the same location ("same" or "different"). In the two *bisensory localization* tasks, participants were asked for either the visual stimulus's location (BV) or the auditory stimulus's location (BA) across interleaved trials.

The five tasks—UV, UA, BC, BV, BA—differed in the response type demanded from participants. We refer to UV and UA collectively as unisensory tasks (which involve only localization), and the other three as bisensory tasks (which involve both explicit or implicit causal inference and localization). In the experiment, the five tasks were presented in four alternating blocks of trials, with BV and BA trials part of the same "bisensory localization" block.

In the UA, BC, BV, and BA tasks, each trial featured the presentation of a sinusoidal wave as the auditory stimulus $s_A$, with its horizontal location in degrees of visual angle (°) sampled uniformly from the discrete values $s_A \in \{0°, \pm5°, \pm10°, \pm15°\}$ (Fig 1B). These locations corresponded to seven physical speakers hidden behind a screen.

In the UV, BC, BV, and BA tasks, each trial featured the presentation of a Gaussian luminance spot as the visual stimulus $s_V$, with its horizontal location sampled from the continuous range $s_V \in [-20°, 20°]$ (Fig 1B). The visual stimulus sampling scheme was uniform in the above range for UV tasks; for BC, BV, and BA tasks, the visual stimulus was uniformly sampled in the above range with 50% probability, and was otherwise at the same location as the auditory stimulus. To explore how human participants integrate visual and auditory information according to their respective precision, the visual stimulus in each trial took on one of three reliability levels ("high", "medium", and "low"), which differed in the width

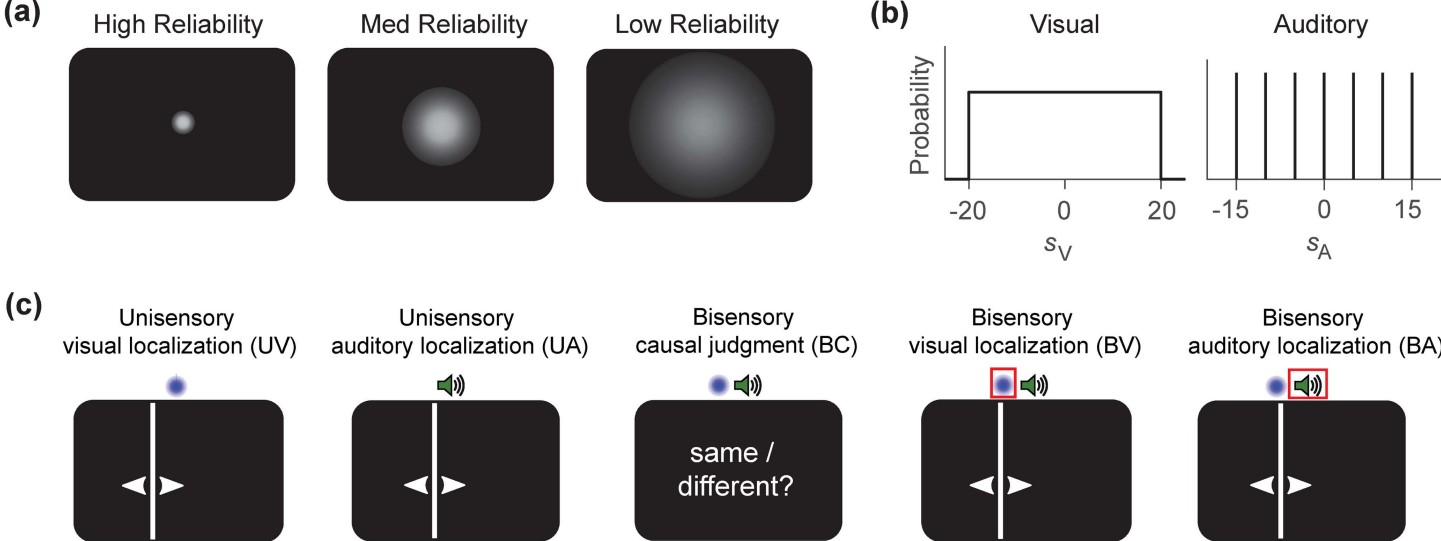

**Fig 1. The five tasks and stimuli in our experiment. (A)** In unisensory localization trials, participants were presented with either a visual stimulus coming in one out of three reliability levels (UV task) or an auditory stimulus (UA task), and were asked to provide a location estimate for the stimulus. In the bisensory trials, participants were simultaneously presented with one visual and one auditory stimulus that either shared or did not share the same location. In the bisensory causal judgment (BC) task, participants were asked to report whether the two stimuli shared the same location; in the bisensory localization tasks, participants were asked to provide a location estimate for either the visual (BV) or auditory (BA) stimulus. **(B)** The visual stimulus's location in each trial is sampled uniformly within $[-20°, 20°]$, while the auditory stimulus's location is sampled from discrete values $\{0°, \pm 5°, \pm 10°, \pm 15°\}$ with equal probability. In bisensory trials, we force the visual stimulus location to be identical to the auditory stimulus ($C = 1$) with probability 0.5 in each trial. **(C)** The visual stimulus is a Gaussian luminance spot that comes in one out of three possible reliability levels (defined by the luminance spot's radius). The open-source speaker icon was adopted from Wikimedia Commons at https://commons.wikimedia.org/wiki/File:Speaker_Icon.svg.

(standard deviation) of the Gaussian luminance spot, with smaller width corresponding to higher reliability level (Fig 1C). The visual stimulus reliability in each trial was randomly chosen with equal probability, $\frac{1}{3}$.

The participant's response in the UV, UA, BV, and BA tasks was a continuous stimulus location estimate $r$ within response range $r \in [-45°, 45°]$ provided via a mouse click. Note that the cursor position on the screen was not reset to any fixed value (e.g., 0°) at the beginning of each trial. The participant's response for the BC task was a discrete same/different judgment $r_C \in \{1, 2\}$ (1 "same", 2 "different"), provided by clicking the corresponding button on the screen.

The experiment consisted of five sessions. Each participant completed 500 UA trials, 500 UV trials, 1000 B (i.e., BV and BA) trials, and 1000 BC trials, for a total of 3000 trials. A total of 15 participants took part in the experiment.

### Exploring the unisensory data

We started by examining the participants' responses in the unisensory tasks (Fig 2).

In Fig 2A, we visualize the empirical response distributions $p(r|s)$ for UV and UA trials constructed from participants' responses. Stimuli were binned for the purpose of visualization (see Methods). While care is needed when examining averages over participants, we can highlight some broad features to investigate further at the participant level.

First, we observe a noticeable difference in the spread of the response distributions as a function of true stimulus location. Response distributions at peripheral stimulus locations are more spread-out, while the response distributions corresponding to straight-ahead stimuli ($s \approx 0°$) are comparably much narrower.

Another prominent feature of the UV responses is the *central tendency*, starkly visible for example in the $s \in [-8.6°, -2.9°]$ and $s \in [2.9°, 8.6°]$ response distributions for the UV low visual reliability condition: the peaks of these two response distributions are pulled towards the center of the horizontal axis ($r = 0°$). Lastly, across rows, the spread

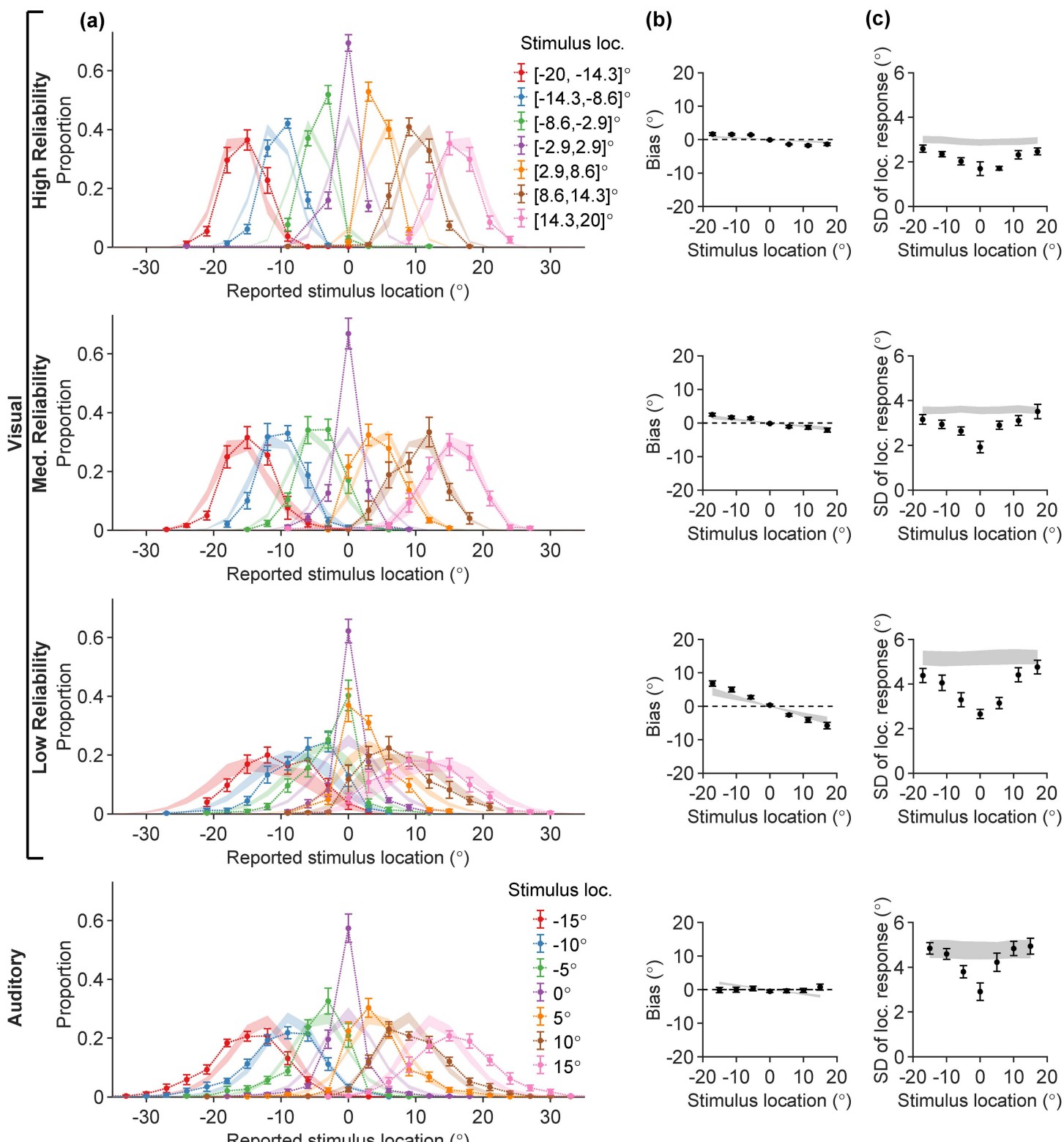

**Fig 2. Behavior in unisensory visual (UV) and auditory (UA) tasks.** Each column shows participants' data (error bars; mean ± SEM across *N* = 15 participants) and model fits (ribbons) for the "vanilla" Bayesian observer model, for different summary statistics. Rows correspond to UV trials (in three

visual reliability levels) and UA trials. **(A)** Full response distributions stratified by true stimulus location into 7 evenly spaced stimulus bins (colors); **(B)** Mean location estimation bias (vertical axis), evaluated separately for each of the 7 stimulus bins (error bars are located at the bin centers along the horizontal axis); **(C)** Standard deviation (SD) of stimulus location estimates (vertical axis), evaluated separately for each of the 7 stimulus bins (horizontal axis).

of UV response distributions increases as the visual reliability decreases, and the spread is generally larger for UA trials compared to the high visual reliability UV trials.

To investigate these features in more detail, we compute summary statistics (mean and standard deviation) of the reported stimulus location. In Fig 2B, we plot the response bias—defined as the mean difference between location estimates and the true stimulus location—for each of the 7 stimulus bins, averaged across participants. For UV, the bias magnitude is larger in the periphery and most prominent in the low-reliability condition, which indicates that the mean location estimates are pulled more strongly towards the center (central tendency). In contrast, we observe no central tendency in the UA task, where location estimates appear on average unbiased.

In Fig 2C, we plot the standard deviation (SD) of stimulus location estimates. The SDs of responses feature a prominent dip at the center of the stimulus range for $s = 0$, and grow larger towards peripheral locations at a sublinear rate.

We attempted to explain the unisensory data by fitting the standard "vanilla" Bayesian observer model. This is a baseline, standard Bayesian observer model commonly used in perceptual estimation [1,10,14,25,26,46]. This model assumes that observers combine noisy measurements with a central prior to yield a posterior estimate which is effectively a linear combination of the measurement and the prior mean, each weighted by their reliability (inverse variance). Formally, this assumes a Gaussian prior and a Gaussian measurement noise which is constant for all stimuli (details in the Models section below). The model fits are shown as ribbons in Fig 2, where the ribbon width corresponds to ± SEM of the model predictions. Overall, in Fig 2A the match between the model-predicted response distributions and the participants' data is poor, as the model fails to correctly capture the changing spread of the responses. This discrepancy is also manifested in the SD plots of Fig 2C, where the model-predicted ribbons fail to capture the dip around $s = 0$. Moreover, the model struggles to explain the lack of central tendency for the UA trials – unless auditory measurements are noiseless, or participants have a nearly flat prior. This drastic failure of the standard Bayesian observer model motivates us to introduce a broader class of Bayesian observer models that deviate from common previous assumptions in the field, with the aim of providing a better explanation of empirical data while remaining consistent with findings from the psychophysical literature.

## Models

To model the data, we considered a family of Bayesian observer models summarized in Fig 3. In the following, we denote by $x \sim p(x)$ a random variable $x$ with probability density function (pdf) given by $p(x)$, and with $\mathcal{N}\left(x; \mu, \sigma^2\right)$ a Gaussian distribution with mean $\mu$ and variance $\sigma^2$. For detailed equations and derivations, see the Methods section. See [10] for an introduction to Bayesian observer modeling in perception.

**Unisensory localization tasks (UV and UA).** To model the UV and UA tasks, we assume the following generative process. The stimulus appears at the true stimulus location $s$ ($s_V$ for the UV task, $s_A$ for the UA task), but the observer has access only to a noisy observation $x \sim p(x|s)$ generated from the measurement distribution $p(x|s)$ ($x_V$ for the UV task, $x_A$ for the UA task, respectively). We assume $p(x|s) = \mathcal{N}\left(x; s, \sigma^2(s)\right)$, a Gaussian distribution centered on $s$ and with standard deviation $\sigma_V(s)$ or $\sigma_A(s)$, which are stimulus-dependent *sensory noise functions*, defined below. For a given $x$, the Bayesian observer employs Bayes' rule to compute a posterior distribution over stimulus location, $p(s|x) \propto p(x|s)p(s)$, where $p(x|s)$ here is the likelihood function (a function of $s$, for a fixed measurement $x$) and $p(s)$ is the observer's prior over the stimulus location, defined below. The observer then internally computes the posterior mean estimate $\hat{s}(x)$. Finally, the reported response $r$ is equal to the observer's estimate corrupted by two noise processes: Gaussian motor noise added to the estimate, and a probability of "lapsing", i.e., giving a uniform random response within the response range

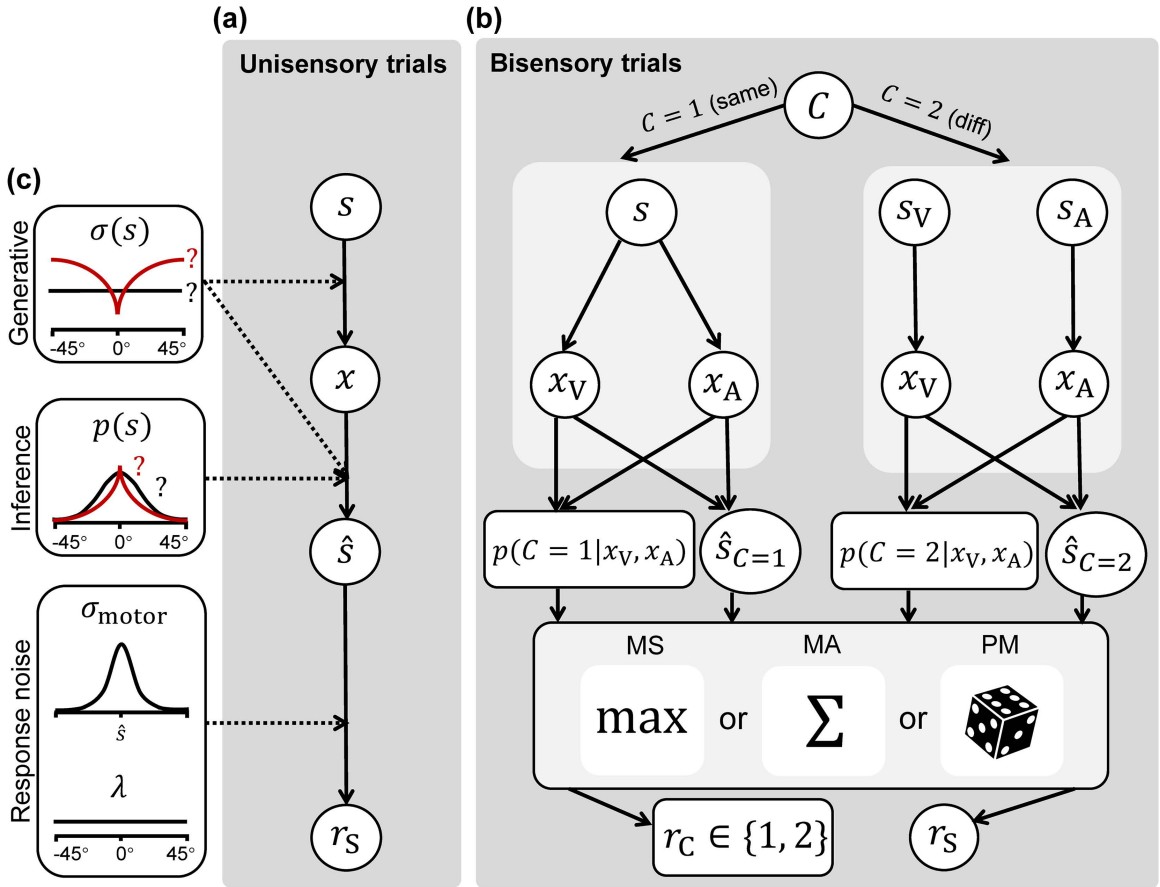

**Fig 3. The generative process and Bayesian observer models for our tasks. (A)** For unisensory tasks UV and UA, observers have access to a noisy measurement $x$ of the stimulus $s$ being presented. They perform Bayesian inference to obtain the posterior mean stimulus value $\hat{s}$, which is corrupted by Gaussian motor noise with SD $\sigma_{motor}$ and occasional, uniformly distributed lapses with lapse rate $\lambda$ to reach the final location estimate response $r_S$, for $S \in \{V, A\}$. **(B)** For bisensory tasks BC, BV, and BA, observers consider two candidate hypotheses—that the auditory or visual stimuli either share ($C=1$) or do not share ($C=2$) the same location—and perform Bayesian inference accordingly. In the BC task, the posterior probabilities of either hypothesis $p(C = 1|x_V, x_A), p(C = 2|x_V, x_A)$ are processed by a causal inference strategy: Model Selection (MS), Model Averaging (MA), or Probability Matching (PM) (MS and MA are equivalent strategies for BC); and corrupted by occasional uniformly distributed lapses to reach a category response $r_C$. In BV and BA tasks, $p(C = i|x_V, x_A)$ for $i = 1, 2$ and the posterior mean location estimates under each hypothesis $\hat{s}_{C=1}, \hat{s}_{C=2}$ are processed by a causal inference strategy and corrupted by Gaussian motor noise and occasional uniformly distributed lapses, ultimately reaching a location estimate response $r_S$. **(C)** In the above process, the shape of the sensory noise function $\sigma(s)$ influences both the generation of noisy observations and Bayesian inference, while the prior over stimulus location $p(s)$ influences the latter. The causal inference strategy is only relevant for bisensory tasks. For simplicity, this figure does not depict further modeling elements: the binary prior over causal inference hypotheses $p_{same}$, the auditory range recalibration parameter $\rho_A$, the rescalings applied to visual sensory noise for medium- and low-reliability trials ($\alpha_{med}, \alpha_{low}$), nor the unisensory-to-bisensory rescaling parameters on sensory noise ($\beta_V, \beta_A$). See text for more details. The open-source dice icon was adapted from Wikimedia Commons at https://commons.wikimedia.org/wiki/File:Dice_simple_flat.svg.

[$-45°, 45°$]. In formula, $p(r|\hat{s}) = (1 - \lambda)\mathcal{N}(r; \hat{s}, \sigma^2_{motor}) + \lambda\frac{1}{L}$ where $\sigma_{motor}$ is the motor noise standard deviation, $\lambda$ the lapse rate, and $L = 90°$ is the width of the response range.

**Bisensory causal judgment task (BC).** For the bisensory tasks BC, BV, and BA, the causal inference strategy specifies how the observer combines the posterior estimates under each hypothesis $C \in \{1, 2\}$, whereby the visual and auditory stimuli either share ($C=1$) or do not share ($C=2$) the same location. We consider three causal inference

strategies commonly used in previous studies: Model Averaging (MA) [1,3,25,26], Model Selection (MS) [1,25,26], and Probability Matching (PM) [25,62,63].

To model the BC task, we consider the two available causal inference hypotheses $C=1$ (same location) and $C=2$ (different locations). We assume the observer uses Bayes' rule to compute the posterior over the binary causal hypothesis $C$, $p(C|x_V, x_A) \propto p(x_V, x_A|C)p(C)$, where $p(C)$ is the discrete prior over the two hypotheses (parameterized by $p_{same} \equiv p(C = 1)$), while the likelihood $p(x_V, x_A|C)$ is computed by marginalizing over all possible true stimulus locations $s = s_V = s_A$ (for $C=1$) or $s_V, s_A$ (for $C=2$). After obtaining posterior probabilities $p(C = 1|x_V, x_A)$ and $p(C = 2|x_V, x_A)$, the observer computes their estimate $\hat{C} \in \{1, 2\}$ according to the model's causal inference strategy. The response $r_c$ is either the hypothesis with the higher posterior probability (MS and MA, which are equivalent in the BC task), or stochastically generated according to the respective posterior probabilities of the two hypotheses (PM). The reported judgment $r_c$ is additionally corrupted by a probability $\lambda$ of "lapsing", in which case the observer simply reports $r_c=1$ or $r_c=2$ with equal probability.

**Bisensory localization tasks (BV and BA).** To model the BV and BA tasks, similarly to the BC task, we assume the observer considers the two distinct causal inference hypotheses $C=1$ and $C=2$ and uses Bayes' rule to calculate the posterior mean of the required stimulus location ($s_V$ for BV, $s_A$ for BA) separately under each hypothesis, that is $\hat{s}_{C=1}$ and $\hat{s}_{C=2}$. Then, according to the causal inference strategy, the observer reports one of the following estimates: **(A)** either one of the two posterior mean values $\hat{s}_{C=i}$ associated with the larger posterior probability $p(C = i|x_V, x_A)$ for $i=1,2$ (MS); **(B)** a linear combination of $\hat{s}_{C=1}$ and $\hat{s}_{C=2}$, weighted by the posterior probabilities of their corresponding hypotheses $p(C = 1|x_V, x_A)$ and $p(C = 2|x_V, x_A)$ (MA); or **(C)** either one between $\hat{s}_{C=1}$ and $\hat{s}_{C=2}$ chosen stochastically with probabilities equal to $p(C = 1|x_V, x_A)$ and $p(C = 2|x_V, x_A)$, respectively (PM). The reported estimate is further corrupted by lapse and motor noise, as per the formulation described above for UV and UA trials.

**Prior.** In all our Bayesian observer models, the prior pdf over stimulus locations $p(s)$ is assumed to be symmetrical with respect to $s=0$ and monotonically decreasing towards the periphery. In this paper, for the prior we consider parametric pdfs (e.g., Gaussian), mixtures of parametric pdfs (e.g., a mixture of Gaussian and Laplace pdfs), and flexible, semiparametric pdfs described later. In all cases, $\theta_{prior}$ is a vector representing the free parameters of such probability distributions. In all tasks, we assume the same prior for visual and auditory stimuli, i.e., $p(s_V) = p(s_A)$. Here, our main interest lies in testing whether prior shapes beyond Gaussian better describe human expectations in perceptual causal inference.

**Sensory noise functions.** The sensory noise function for vision and audition are denoted $\sigma_V(s; \theta_V)$ and $\sigma_A(s; \theta_A)$ respectively, parameterized by free parameter vectors $\theta_V, \theta_A$. The notations $\sigma(s)$ and $\theta_\sigma$, without specifying a sensory modality, are used when discussing properties that hold for both vision and audition. We assume that the sensory noise functions are symmetrical with respect to $s=0$ and monotonically increasing towards the periphery. While the latter assumption might stop holding for large eccentricities, stimuli in the experiment are confined in the $[-20°, 20°]$ range. All our models assume a Gaussian measurement distribution $p(x|s) = \mathcal{N}(x; s, \sigma(s))$. Here, our aim is testing realistic observer models where $\sigma(s)$ is not a constant function over $s$ (i.e., heteroskedastic sensory noise). We will explore the shape of such heteroskedastic $\sigma(s)$ functions, and compare their ability to describe human behavior to the more commonly used homoskedastic sensory noise functions, where $\sigma(s)$ is constant over $s$.

**Additional details.** Given the presence of three visual reliability levels, we define the visual sensory noise function $\sigma_V(s)$ with respect to the high reliability (low noise) level for the unisensory tasks. We assume that the entire sensory noise function is scaled by multiplicative noise factors $\alpha_{med}$ and $\alpha_{low}$ for the medium- and low-reliability trials, respectively, i.e., the overall noise profile is the same across visual reliabilities, but with different scaling factors.

We also considered that during bimodal trials, the diversion of attentional resources across sensory modalities may result in larger sensory noise compared to unisensory tasks [64,65]. To account for this potential effect, we included unisensory-to-bisensory noise rescaling parameters $\beta_V$ and $\beta_A$, which are applied as multiplicative factors to the sensory noise functions $\sigma_V(s), \sigma_A(s)$, respectively, during bisensory tasks only.

Finally, due to the difference in the ranges of visual and auditory stimuli used in the experiment (–20° to 20° for visual stimuli; –15° to 15° for auditory stimuli), we also considered models that include an auditory range recalibration parameter $\rho_A$. This parameter effectively "stretches" the auditory noisy measurement $x_A$ before being processed by the observer. We introduced this parameter motivated by the well-known phenomenon of sensory recalibration in the presence of a consistent mismatch between modalities [49,66–74], and its importance will be discussed and validated empirically in the next section.

**Model summary.**  Overall, our Bayesian observer models mainly differ in three components: (A) the sensory noise functions for vision and audition $\sigma_V(s; \theta_V)$, $\sigma_A(s; \theta_A)$; (B) the prior over stimulus location $p(s; \theta_{prior})$; (C) the causal inference strategy (MA, MS, or PM), which is relevant for bisensory tasks only. The free parameter vectors $\theta_V, \theta_A, \theta_{prior}$ control the shapes of the sensory noise functions and prior distribution, respectively, and belong to the model's free parameters.

Hence, each of our models consists of the following free parameters:

$$(\theta_V, \theta_A, \theta_{prior}, \lambda, \sigma_{motor}, \alpha_{med}, \alpha_{low}, p_{same}, \beta_V, \beta_A),$$

where $(p_{same}, \beta_V, \beta_A)$ are only relevant when fitting bisensory tasks. Models that assume auditory range recalibration have the additional recalibration parameter $\rho_A$.

**Vanilla Bayesian observer model.**  The standard or "vanilla" Bayesian observer model shown previously in Fig 2 is characterized by a simple parametric likelihood and prior, assuming a constant ("homoskedastic") sensory noise function $\sigma(s) = \sigma_0$ and a Gaussian prior pdf $p(s) = \mathcal{N}(0, \sigma_s^2)$.

## Auditory range recalibration

The "vanilla" Bayesian observer model fails to capture a key difference between human UV and UA data: UV data show a clear central tendency at all reliabilities, which is absent in UA trials (Fig 2B), despite auditory localization variability being comparable to low-reliability visual stimuli (Fig 2C). This is puzzling from a Bayesian observer perspective, as less reliable cues are expected to exhibit a stronger attraction towards a central prior [75–78].

Here, we propose that the central tendency in participants' responses is cancelled out by an opposing tendency arising from the difference in range between visual and auditory stimuli in the experiment. As a reminder, visual stimuli are drawn from the range $[-20, 20]°$, whereas auditory stimuli are drawn from $[-15, 15]°$. This difference may have induced in the participants a remapping of their auditory observations to the wider visual stimulus range via a multiplicative gain [66–69]. We assume that the considered recalibration is from auditory to visual since vision included high-reliability trials and, crucially, the localization responses were always given via a visual cursor. To address this possibility, we fitted the vanilla observer model with an additional auditory range recalibration parameter $\rho_A$, which represents the multiplicative gain applied to the auditory measurements. If observers were exactly remapping the auditory stimulus range ($\Delta s_A = 30°$) to the visual stimulus range ($\Delta s_V = 40°$), we would expect to find $\rho_A = \frac{4}{3}$. If no remapping were taking place, we should find $\rho_A = 1$.

We fitted both a model in which $\rho_A \equiv \frac{4}{3}$, corresponding to "full remapping" of the auditory range to the visual range, and a model in which $\rho_A$ is left as a free parameter.

Quantitative model comparison results show that both models achieve substantially lower (better) AIC and BIC scores than the original vanilla model. In particular, the free recalibration parameter model is better than both other vanilla models (details are available in S1 Appendix Section B). Importantly, while this result suggests that not all observers perform an exact "full remapping" and the real process is likely more complex, we find that the group average of the auditory range recalibration parameter is $\rho_A = 1.40 \pm 0.07$ (mean ± SEM), which is remarkably close to and statistically indistinguishable from the expected value of $\rho_A \approx 1.33$ for a full remapping (two-sided t-test, $t(14) = 0.92$, $p = 0.37$). Overall, these results

suggest that observers are recalibrating their auditory range to approximately match the visual range, validating our hypothesis. This justifies the inclusion of the auditory range recalibration parameter $\rho_A$ in our later models.

While accounting for the auditory range recalibration improved our model fits for the vanilla observer model, the fits still present the same general issues highlighted previously—failing to account for the strong central tendency and central SD dip as seen in humans (see Fig F in S1 Appendix Section B). For this reason, we next consider a different modeling approach.

### Semiparametric modeling of prior and noise

Our working hypothesis is that vanilla observer models fail to capture our data due to (A) the shape of the sensory noise function and (B) the shape of the prior. The default choices—constant noise and a Gaussian prior—may be excessively simple assumptions. First, we know from the psychophysical literature that the noise of both vision and audition increases with eccentricity [79–83]. Second, while a Gaussian prior is often a good approximation, observers often exhibit biases that may arise from more complex environmental statistics, such as a diffuse prior which "peaks" at more likely locations, such as cardinal orientations [57] or, as in our case, straight ahead [47,84].

However, moving beyond the single-Gaussian prior, constant-noise case is nontrivial. In principle, we could propose a discrete set of parametric forms for the prior and the noise functions, and perform a *factorial model comparison* among all prior and noise combinations to determine which one describes the data best [40,58,85]. However, such an approach has two major downsides. First, a factorial analysis is limited to the choice of priors and noise functions included a priori in the comparison. While we could devise a variety of ad hoc shapes for the prior and noise, there is no simple principled way to decide which shapes we should consider. Second, on the practical side, a factorial comparison wide enough to include a broad choice of shapes quickly leads to a combinatorial explosion, with hundreds of models to consider [85,86].

Instead, we propose to estimate the prior and noise function shapes—separately for vision and audition—through the data, via a "semiparametric" approach. We use this term to mean that we parameterize a shape (the pdf or the noise function) via the values it takes on a pre-determined grid, and then we smoothly interpolate. This approach is more flexible than a typical parametric approach in that we can approximate a vast number of smooth functions with relatively weak assumptions. In the rest of this section, we first describe how we applied the semiparametric modeling approach to the unisensory data, so as to estimate the shapes of the prior and noise functions for each participant. Then, we describe how we used the estimated semiparametric shapes for fitting the bisensory data.

### Semiparametric estimation of prior and noise shapes from unisensory data

We consider here semiparametric representations of the shapes of the sensory noise functions $\sigma_V(s), \sigma_A(s)$ as well as the prior pdf $p(s)$. For the semiparametric representation, we selected 12 locations $[0°, 0.1°, 0.3°, \ldots, 20°, 45°]$ along the non-negative screen range ($0°$ to $45°$), denoted as "pivots". Each semiparametric $\sigma(s)$ and $p(s)$ is then defined via their function values at these 12 pivots. We included the only constraint that priors are monotonically non-increasing in $|s|$, and noise shapes are monotonically non-decreasing in $|s|$, within the entire stimulus range of $[-45°, 45°]$. We obtain intermediate values of each function via smooth interpolation (using cubic splines) of their values at the pivots, and negative values by assuming mirror symmetry about the x-axis. The function values at the pivots, $\theta_V$, $\theta_A$, and $\theta_{prior}$, are free parameters. Using these semiparametric prior and noise shapes, we fitted the observer models described in Models to each participant's unisensory data.

The semiparametrically estimated $\sigma_V(s), \sigma_A(s)$ and $p(s)$ functions for each participant are shown in Fig 4. The $\sigma(s)$ shapes show a steep dip around 0 degrees and plateau towards the periphery. The $p(s)$ shapes show a sharp peak in the center with a longer, broader tail, a shape akin to a mixture of a narrow and a broad component.

To validate that the semiparametric fits explain our participants' data well, we visualized the response distributions predicted by our fitted model on the unisensory data in Fig 5. Overall, there is a good match between the model-predicted

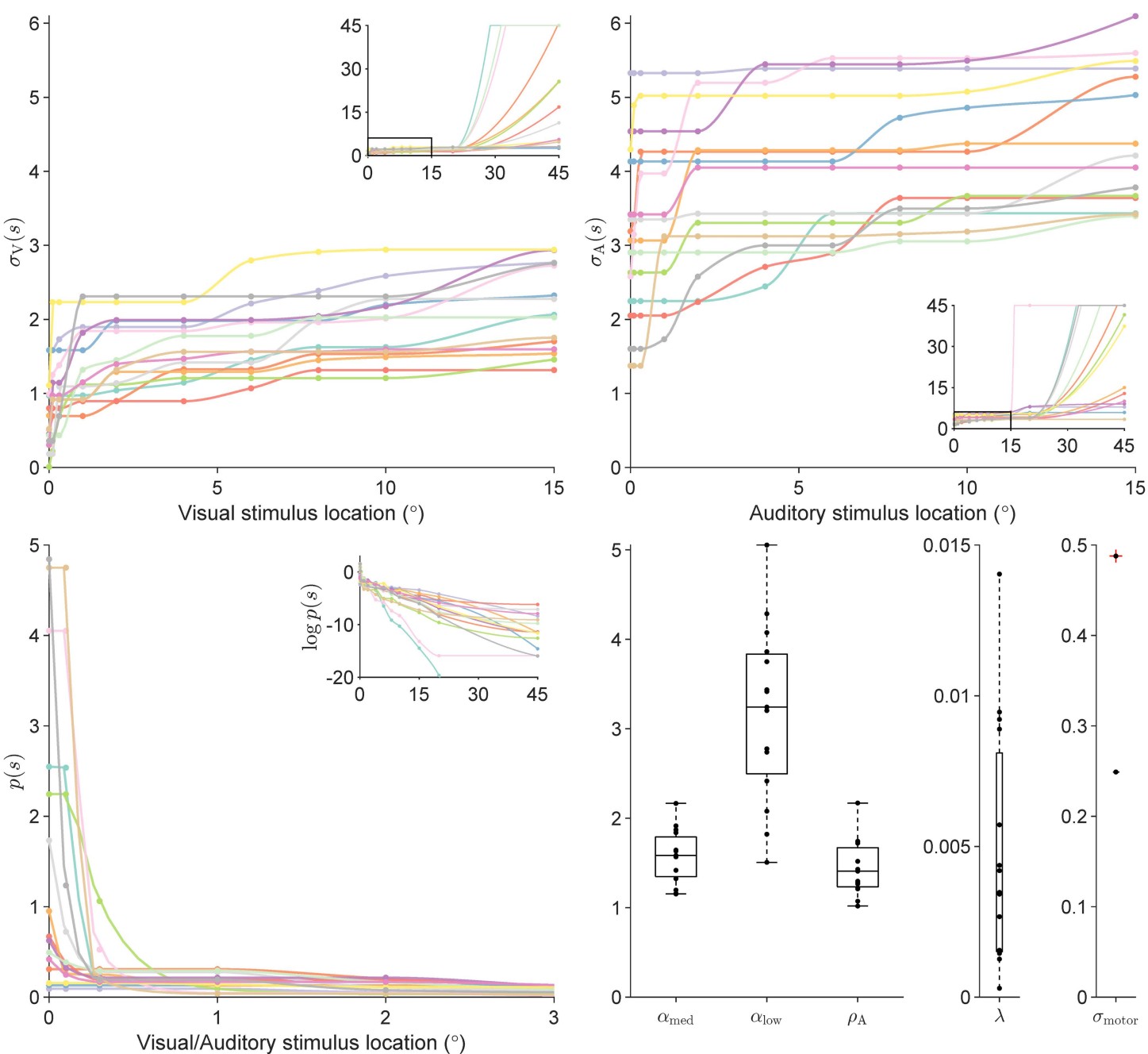

**Fig 4. Semiparametric fits on unisensory data.** Different colors denote different human participants. **(A)** Fitted $\sigma_V(s)$ shapes for each participant, where points denote the pivot locations and their corresponding function values. Inner plot: the same plot over the full stimulus range $s \in [0°, 45°]$; the smaller stimulus range used in the outer plot is denoted by a black rectangle in the inner plot. **(B)** Fitted $\sigma_A(s)$ shapes. Inner plot: the same plot over the full stimulus range $s \in [0°, 45°]$. **(C)** Fitted $p(s)$ shapes. Inner plot: $\log p(s)$ over the full stimulus range $s \in [0°, 45°]$. **(D)** Other fitted parameters. Box plots denote median and interquartile range, and whiskers the full range, with dots being individual participants.

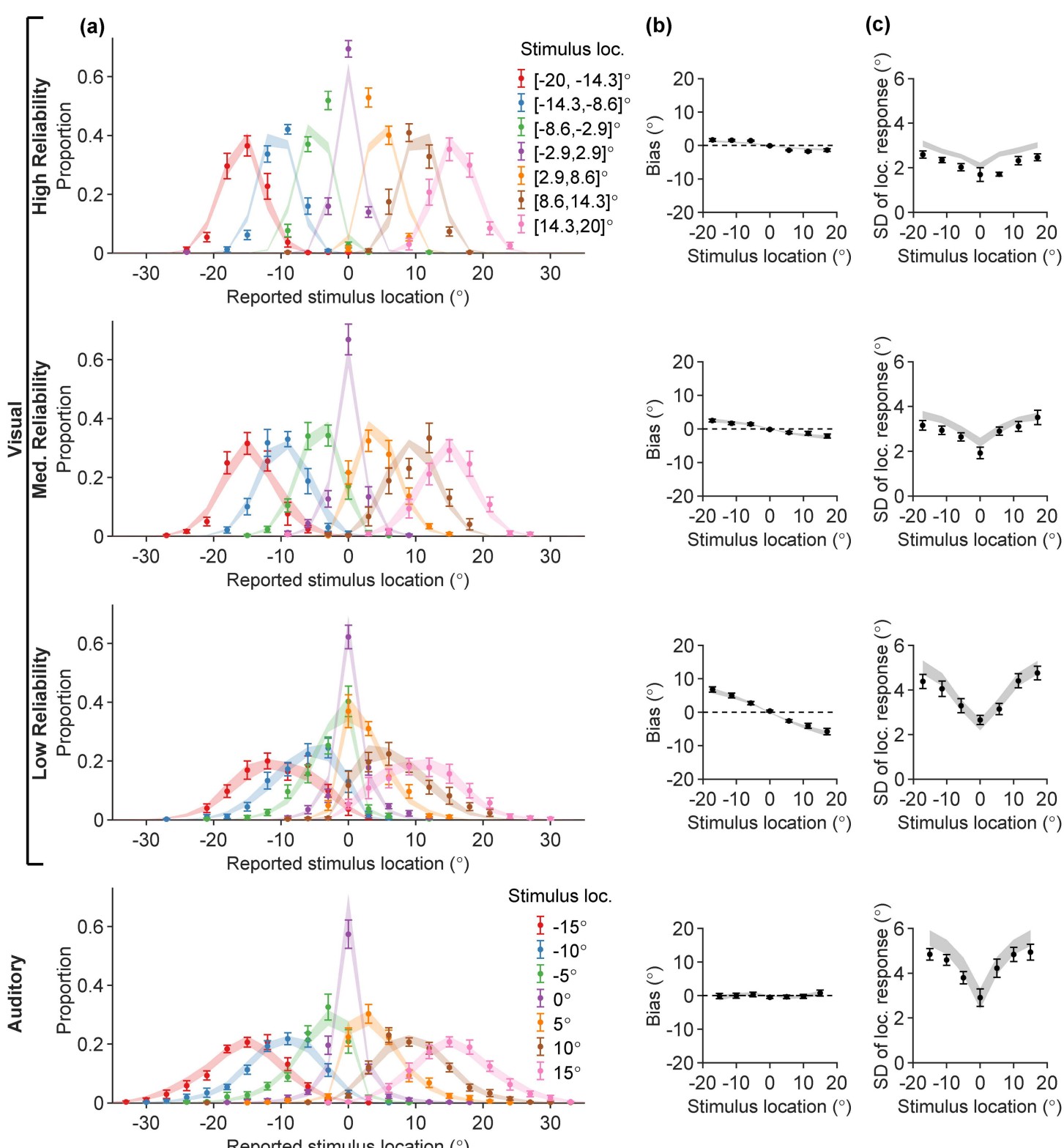

**Fig 5. The semiparametric model fitted jointly on UV and UA data for all participants.** Subplot notations are identical to Fig 2.

response distributions and the human response distributions. Importantly, the combination of a sharply peaked $p(s)$, heteroskedastic sensory noise $\sigma(s)$, and auditory range recalibration $\rho_A$ has allowed the model to capture the behavioral signatures we observed in this task—the prior and sensory noise function induce strong central tendencies in UV, while auditory range recalibration maintains unbiased UA estimates amid the prior.

**Lifting the semiparametric fits to all tasks**

We now investigate the generalizability of the semiparametric $\sigma(s)$ and $p(s)$ functions estimated from unisensory data, by directly "lifting" these fits to explain the data of all tasks.

Specifically, for each participant we perform new model fits on all the data by letting the auxiliary parameters $(\alpha_{\text{med}}, \alpha_{\text{low}}, \lambda, \sigma_{\text{motor}}, \rho_A, \beta_V, \beta_A)$ free to vary, while the key sensory noise function and prior parameters $\theta_V, \theta_A, \theta_{\text{prior}}$ are lifted directly from the earlier semiparametric fits on unisensory data (Fig 4). We call these new models *lifted-semiparametric models*. To give the lifted prior shape a modicum of flexibility, we introduced a single shape parameter $\gamma$, but otherwise we are under the strong assumption that the prior and noise shapes transfer exactly from unisensory to bisensory trials; see Methods for details.

Since we are working with bisensory tasks, there are three possible distinct causal inference strategies (MS, MA, PM), leading to three sets of distinct model fits. In the main text, we discuss the best lifted-semiparametric model according to AIC and BIC, LiftedSemiparam-PM. Full model comparison results are available in Fig 9 and S1 Appendix Section B.

Moreover, given that the lifted-semiparametric model carries over the sensory noise and prior shapes from the unisensory fits, the lifted-semiparametric predictions for unisensory data are essentially the same as in Fig 5 and are shown in Fig L in S1 Appendix Section C; here we focus on the predictions for bisensory data.

We first inspected model fits on the BC task. We visualize the human BC data (Fig 6A; error bars) as well as the predictions of the model (ribbons), depicting the frequency of responding $C = 1$ ("same") as a function of stimulus location disparity ($s_A - s_V$). The visualization is stratified according to the visual stimulus's reliability level and the absolute value of the sum of the stimulus locations, $|s_A + s_V|$. While the model captures the proportion of $r_c = 1$ in humans well across most stimulus disparity magnitudes, it tends to overestimate this proportion as the stimulus disparity magnitude grows large.

Similarly, the lifted-semiparametric model captures the human BV data consistently well. We visualize the human BV data (Fig 6B; error bars) as well as the predictions of the model (ribbons), the mean estimation bias ($\hat{s}_V - s_V$) as a function of stimulus location disparity ($s_A - s_V$). The visualization is stratified according to the visual stimulus's reliability level and whether the sum of stimuli locations ($s_A + s_V$) is to the left, center, or right. The visualization shows that our model replicates the dependency of human estimation bias on stimulus location disparity, in which the bias asymptotes at large disparities instead of scaling linearly. This nonlinearity suggests that people reduce their weight for the auditory stimulus when stimulus disparity is very large, as prescribed by optimal causal inference. Second, for both humans and model predictions, the slopes of mean estimated bias over stimulus disparity are shallower for higher visual reliabilities. This slope difference indicates that human participants are weighing visual and auditory according to their reliability, a characteristic of Bayesian cue integration.

Fig 6B also visualizes the human BA data (error bars) as well as the predictions of the model (ribbons). The visualization process is identical to BV data, with the only difference being that the mean estimate bias is computed for auditory stimulus as ($\hat{s}_A - s_A$). The lifted-semiparametric model captures the human BA data well for small stimulus disparity magnitudes and the center ($s_A + s_V$) bin, only underestimating the bias for periphery bins and extreme stimulus disparity magnitudes. For both human data and model predictions, we still observe nonlinearity in the mean estimation bias versus stimulus disparity relationship indicative of causal inference, but the slope difference across visual reliability levels is less pronounced.

The unisensory-to-bisensory noise rescaling parameters are fitted as $\beta_V$ = 1.15 ± 0.03 and $\beta_A$ = 1.58 ± 0.09 respectively (mean ± SEM across participants). Both values are significantly greater than 1 (one-sided t-tests: $\beta_V$: $t(14)$=4.72,

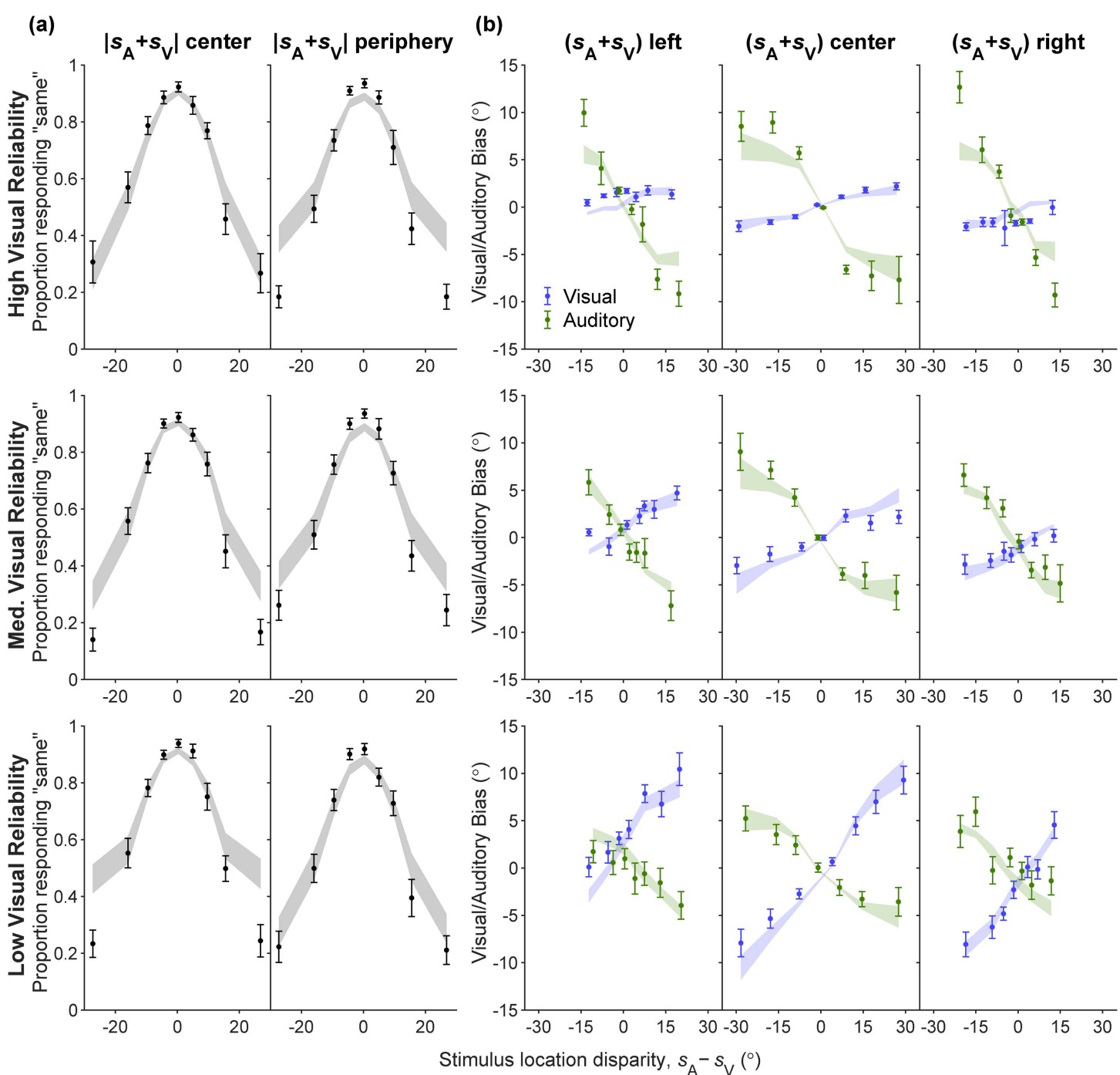

**Fig 6. The lifted-semiparametric fits on all tasks, assuming the PM causal inference strategy, visualized on BC, BV, and BA data.** All response distributions are constructed from data (error bars) and model (ribbons) predictive samples; mean±SEM across participants. Rows correspond to different visual reliability levels. **(A)** BC response distributions, denoting the relative frequency of responding that the two stimuli share the same source ("same" responses), as a function of binned true stimulus location disparities. Columns are stratified into 'center' and 'periphery' based on the sum of the two true stimuli locations (absolute value below or above the corresponding median across all BC trials). **(B)** BV (blue) and BA (green) response distributions, denoting the mean estimation bias as a function of binned true stimulus location disparities. Columns are stratified into 'left', 'center' and 'right', based on the sum of the two true stimuli locations (25th and 75th percentiles across all BV or all BA trials).

$p = 1.64 \times 10^{-4}$; $\beta_A$: $t(14)=6.61$, $p = 5.85 \times 10^{-6}$). Thus, sensory noise in the bisensory task is substantially higher than in the unisensory task.

Overall, the LiftedSemiparam-PM model captures human behavior well across all five tasks. These results confirm that our semiparametrically estimated sensory noise and prior functions not only explain unisensory localization—which they were fitted on—, but also generalize remarkably well to bisensory localization and causal inference.

## Distilled parametric shapes for priors and noise

While powerful and flexible, our semiparametric approach for inferring the shapes of the prior and of the noise function has some important limitations. Due to the large amount of parameters, this approach can be computationally cumbersome, at risk of overfitting, and the results can be hard to summarize or generalize to other experiments. Conversely, parametric approaches with a limited number of interpretable parameters do not have such issues, or at least have them to a lesser extent.

In this section, we propose a "distillation" approach, in that we design a small set of simple parametric shapes for the prior and the noise function inspired by the shapes obtained by the semiparametric approach. We show that these parametric shapes perform closely to – or even better than – the full semiparametric approach, while providing models that are easier to fit and to be used in other studies.

**Unisensory data fits.** We developed parametric models "distilled" from the results of our semiparametric model fit (Fig 4), and fitted such parametric models to the unisensory data (UV and UA), individually for each participant. These fits serve to confirm that our distilled parametric model families provide good descriptions for unisensory localization data, with model comparison metrics comparable to – and even improving over – the much more flexible semiparametric model.

The four distilled parametric models are: Const-GaussianLaplace, Exp-SingleGaussian, Exp-GaussianLaplace, and Exp-TwoGaussians; where the first part of each model name denotes the parametric noise shape (constant, 'Const', or exponential, 'Exp') and the second part denotes the parametric shape for the prior, explained below. We have chosen these four models because they are a minimal combination of shapes inspired by the reconstructed semiparametric noise and prior shapes, $\sigma(s)$ and $p(s)$ (Fig 4), paired with previous assumptions in the field (homoskedastic sensory noise and Gaussian prior). Through these choices, we aim to confirm our hypothesis that heteroskedastic sensory noise functions and non-Gaussian priors both contribute to explaining our human perception data. The results of these unisensory fits will form the foundations for subsequent bisensory fits, which involve both perception and causal inference.

Regarding the sensory noise function, the Exponential (Exp) model assumes $\sigma(s) = \sigma_0 + k_1(1 - \exp(-k_2|s|))$, parameterized by $\theta_\sigma = (\sigma_0, k_1, k_2)$, whereas the Constant (Const) noise model simply assumes $\sigma(s) = \sigma_0$. We chose this exponential sensory noise function motivated by the fitted semiparametric $\sigma(s)$ shapes in Fig 4, which dip steeply towards the spatial center and plateau towards the peripheries. The choice of this sensory noise function form will also be retroactively justified by the quantitative results of the Exp model fits. As for the prior distribution family, the GaussianLaplace and TwoGaussians models have mixture priors, composed of a Gaussian and a Laplace distribution and two Gaussian distributions, respectively, all centered in $0°$ and with three prior parameters $\theta_{prior}$ corresponding to the width of the two mixture components and the relative weight of the two components. Conversely, SingleGaussian is a single zero-mean Gaussian distribution.

In terms of quantitative model comparison, the Exp-GaussianLaplace, Exp-TwoGaussians, and Const-GaussianLaplace models perform in this order but overall similarly well, as shown in Fig 7 for BIC, with similar results for AIC (S1 Appendix Section B). All three models capture unisensory human data similarly well. We visualize the response distributions of the Exp-GaussianLaplace model, the best model in the comparison (Fig 8A-8C; see S1 Appendix Section B for visualizations of the other models). The Exp-GaussianLaplace model captures not only group-level statistics, but also behavior at the individual level, as shown in S1 Appendix Section D.

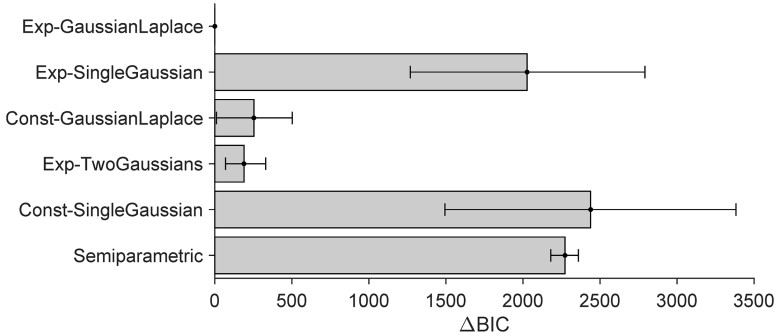

**Fig 7. Model comparison results (BIC) for parametric models fitted on UV+UA tasks data.** The bars report difference in BIC between the model and the best-performing model (Exp-GaussianLaplace), with higher values for worse-fitting models. Error bars are bootstrapped 95% CI.

Instead, the fourth model in our analysis, the Exp-SingleGaussian model, performs both quantitatively and qualitatively worse, failing to capture key features seen in the data, such as the peak at $s = 0°$ response distribution, the correct central tendency in the vicinity of the peak, and the dip in SD around the center compared to the human data (see S1 Appendix Section B). Comparing the performance of Exp-SingleGaussian and Const-GaussianLaplace model, it seems that the presence of a mixture prior is more important than an eccentricity-dependent sensory noise function for explaining human unisensory localization.

From the model comparison (Fig 7) we also see that all distilled models, and particularly the best ones, perform orders of magnitude better than the traditional "vanilla" Gaussian observer model discussed earlier (Const-SingleGaussian, in this notation), with differences of thousands of points of BIC scores (same for AIC, S1 Appendix Section B). Perhaps more surprisingly, we also find that the semiparametric model, the inspiration of our distilled models, performs substantially worse than the distilled parametric models in terms of comparison metrics (both BIC and AIC). This result is due to the penalty that BIC and AIC assign to model complexity, represented by the number of model parameters. In fact, we find that the semiparametric model achieves slightly higher log-likelihood than the distilled models (see S1 Appendix Section B), indicating a better raw match to the data, but the small improvement is not enough to justify the number of parameters.

In conclusion, these findings combined validate our distilled parametric models as a parsimonious and effective description of noise and prior shapes, $\sigma(s)$ and $p(s)$, in our unisensory localization data. Our results confirm that a simple Gaussian prior alone cannot explain the unisensory data, and the most likely explanation appears to be a prior akin to a Gaussian-Laplace mixture combined with a non-constant sensory noise model, here captured by the Exponential noise shape.

**All-tasks fits.** Having confirmed the validity of our approach on the unisensory tasks, we proceeded to fit each of the four distilled parametric models jointly to all tasks (UV, UA, BC, BV, and BA), individually for each participant. For each model we consider distinct causal inference strategies (MS, MA, PM), leading to a total of 12 parametric models (4 base distilled models × 3 causal inference strategies).

In terms of causal inference strategy, the model comparison found *probability matching* (PM) to generally outperform other strategies, with *model averaging* (MA) being a close second, often on par with PM. Instead, *model selection* (MS) is consistently the strategy least supported by the data [25,87]. Importantly, variations in the base parametric function families for priors and noise lead to substantially greater AIC and BIC disparities compared to variations in causal inference strategy. For this reason, for clarity and without loss of generality, we focus here on models that use the PM causal strategy, with full results in S1 Appendix Section B.

The model comparison results of all-tasks fits are shown in Fig 9 for BIC (see S1 Appendix Section B for AIC). Model comparison confirms the best-performing distilled parametric model as Exp-GaussianLaplace (equally with PM or MA

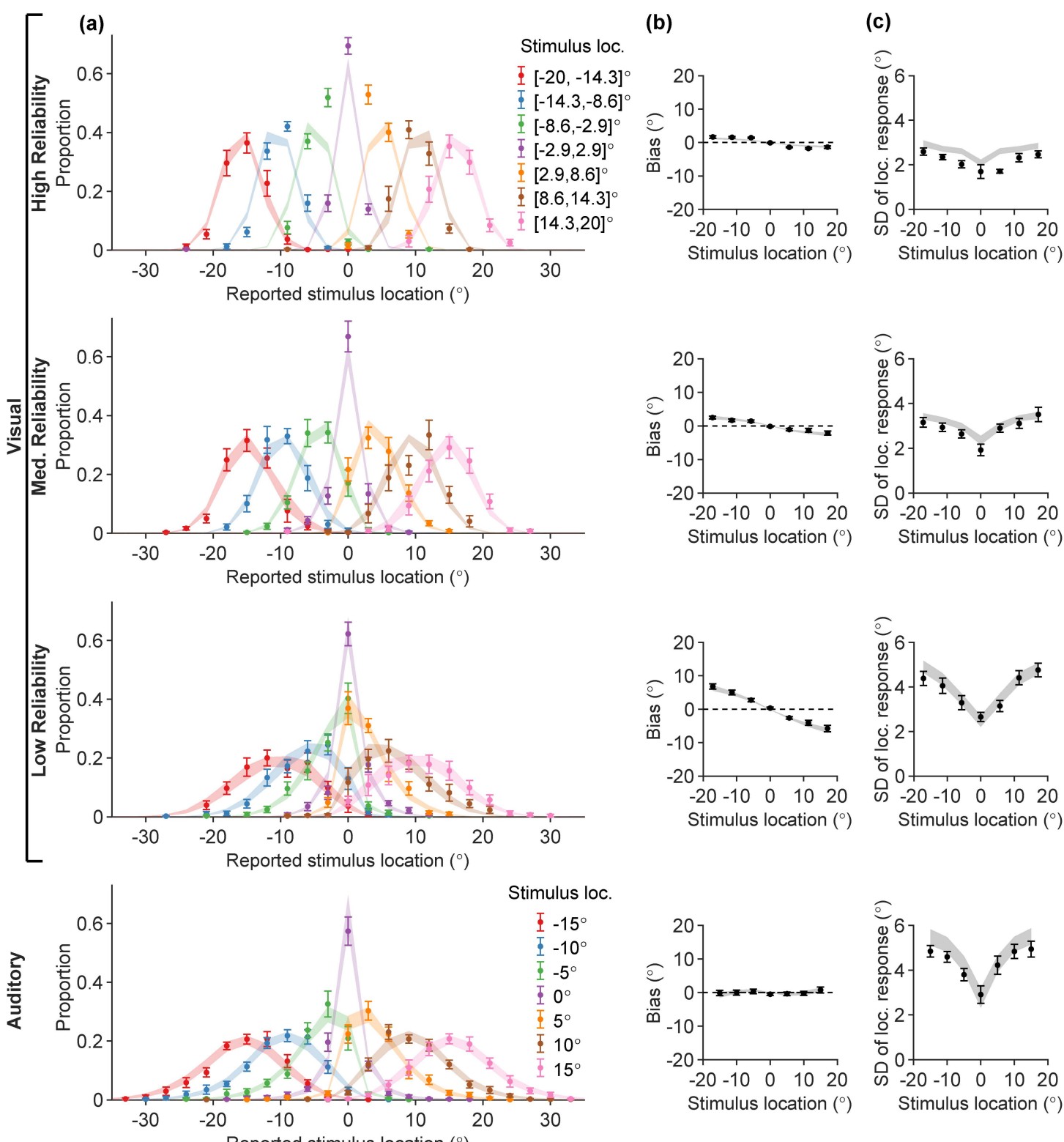

**Fig 8. The Exp-GaussianLaplace parametric model fitted jointly on UV and UA data for all participants, in which the auditory range recalibration factor is a free parameter.** The layout and color schemes are identical to Fig 5.

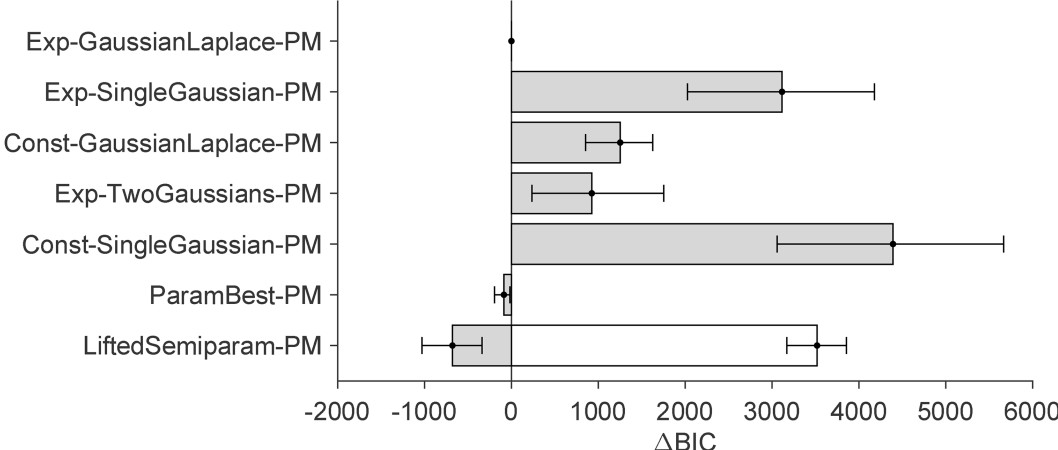

**Fig 9. Model comparison results (BIC) for Probability Matching (PM) parametric models fitted on all tasks.** For the LiftedSemiparam-PM model, many of its parameters were fitted in the earlier semiparametric fits. Its BIC score including such parameters is shown in white, and the BIC score excluding them is shown in gray.

strategies, as mentioned earlier), in agreement with our unisensory-only fits. Exp-TwoGaussians and Const-GaussianLaplace, which performed similarly well in the unisensory fits, are now substantially worse. Exp-SingleGaussian remains the worst-fitting model of the four. Unsurprisingly, the best model majorly outperforms the "vanilla" model (const-SingleGaussian), irrespective of causal inference strategy.

The best distilled parametric model fit, Exp-GaussianLaplace-PM, is visualized in Figs 10-11. The fitted model has parameter values $\rho_A$ = 1.44 ± 0.08, $\beta_V$ = 1.12 ± 0.02, $\beta_A$ = 1.57 ± 0.09, which are quantitatively similar to previous unisensory fits as well as all-tasks lifted-semiparametric fits. The model fit visualizations are of similar quality to LiftedSemiparam-PM (Fig 6), even outperforming the latter in accounting for some details, such as the peripheral BC data (Fig 11A). The BV and BA data are captured similarly well (Fig 11B).

To illustrate that the fitted Exp-GausssianLaplace-PM model can capture not only group-level statistics but also behavior on the individual level, we show its model predictions for a representative participant in Fig 12. Model visualizations for all participants can be found in S1 Appendix Section D. The Exp-GaussianLaplace-PM model matches the participant's data well across the multiple tasks.

To consider the possibility that different participants are best explained by different $\sigma(s), p(s)$ function families, we also took the best distilled parametric model for each participant and computed the ParametricBest set of NLL, AIC, and BIC summed across participants (see Fig 9 for BIC, and S1 Appendix Section B for the other metrics). The ParametricBest model comparison metrics are still on par with the best among all distilled parametric models, indicating that our participants' responses are well explained by a homogeneous family of sensory noise functions and priors.

Finally, we compared our best distilled models to the "lifted" semiparametric models. We found that under a naive model comparison, the lifted-semiparametric models seemingly perform better (according to BIC) than Exp-GaussianLaplace (Fig 9). Conversely, performance is similar in terms of AIC, and the best parametric models perform better in terms of log-likelihood (see S1 Appendix Section B). This mismatch is explained by the way complexity is approximated in BIC (and AIC), merely by counting the number of free parameters. The key issue here is that the *lifted* semiparametric models freeze the parametric shapes of prior and noise after having fitted them to the unisensory data, so these parameters do not count towards model complexity. However, this naive approach gives the lifted SemiParametric fits an unfair advantage – if nothing else, because these fits are 'double dipping' by first fixing a large number of the parameters on the unisensory data. If we apply a simple correction to the number of actual model parameters by also including the

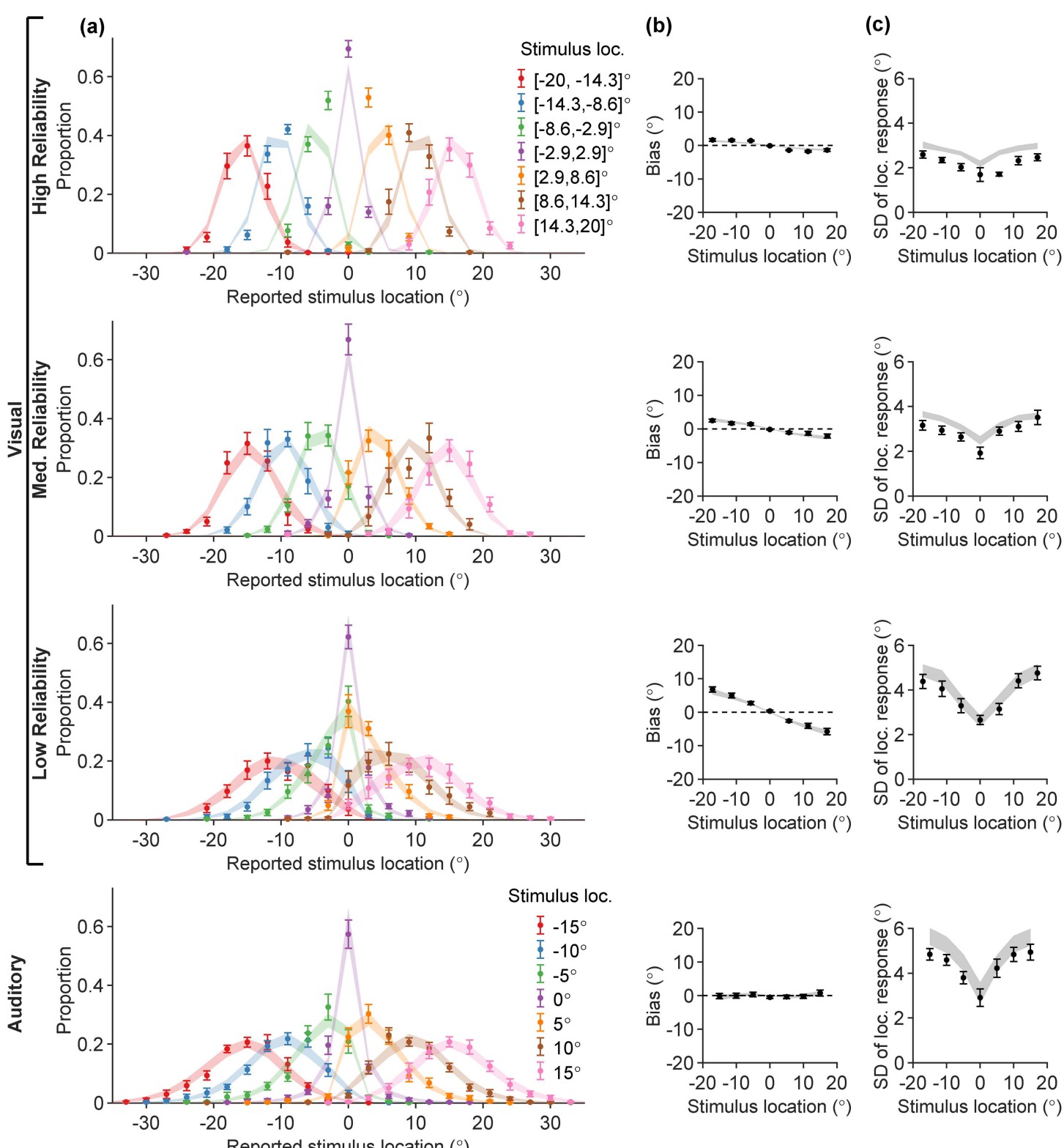

**Fig 10. The Exp-GaussianLaplace-PM parametric model fitted on all tasks, visualized on UV+UA data.** UV+UA response distribution legends are identical to Fig 2.

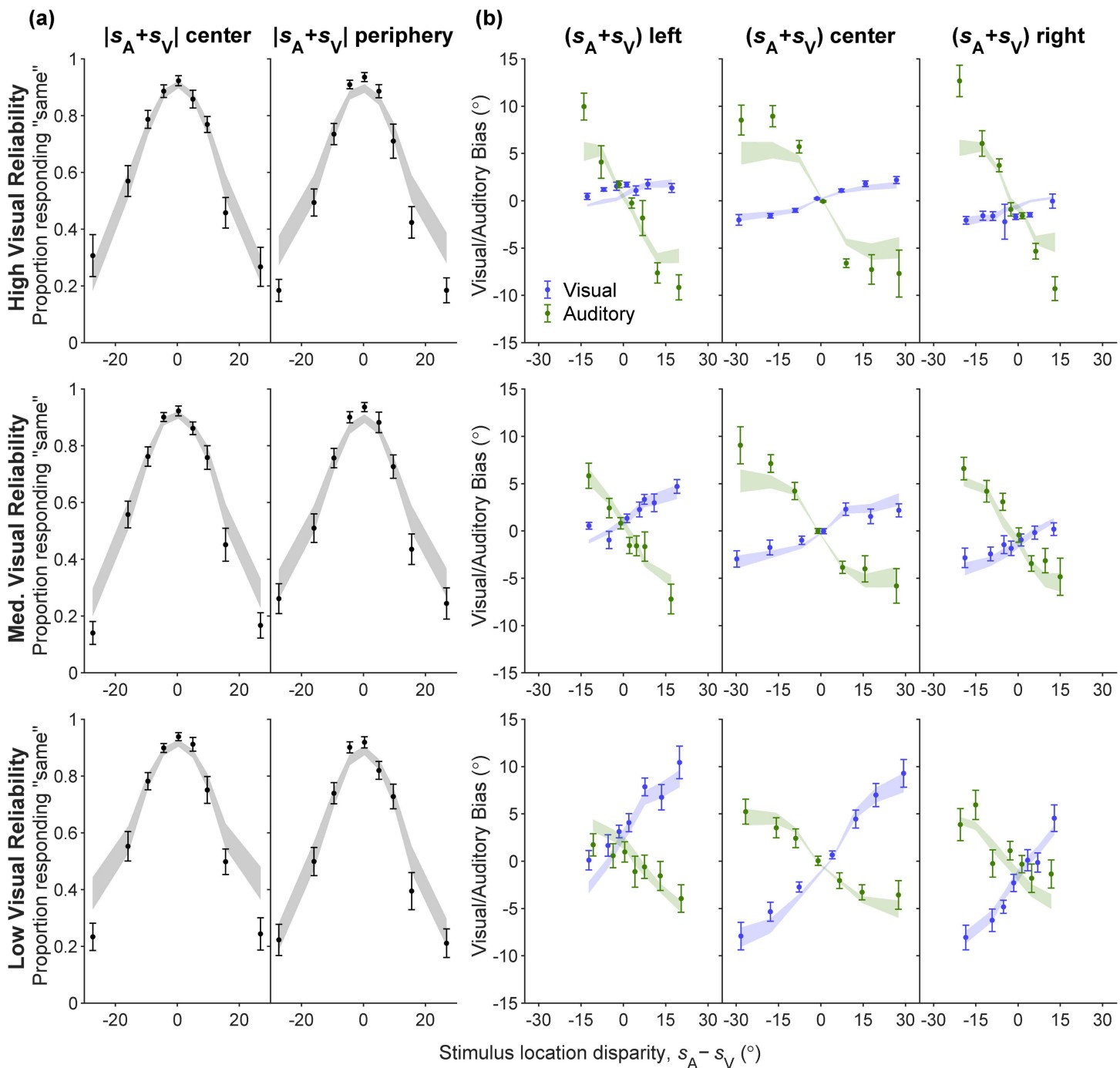

**Fig 11. The Exp-GaussianLaplace-PM parametric model fitted on all tasks, visualized on BC, BV, and BA data.** Response distribution notations are identical to Fig 6.

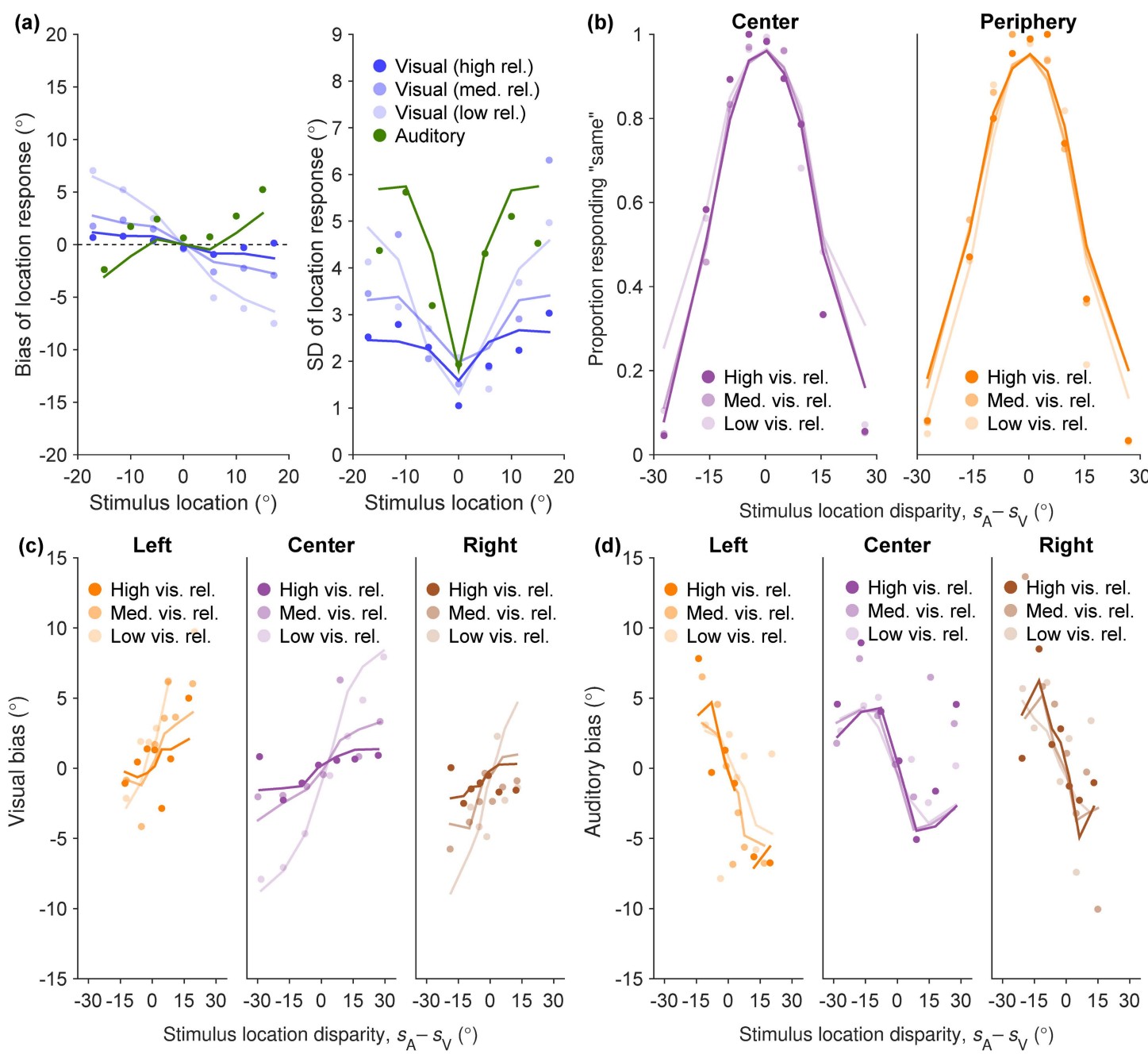

**Fig 12. Responses of one example participant and their Exp-GaussianLaplace parametric model fitted jointly on UV and UA data. (A)** UV and UA data, bias and SD of responses. **(B)** BC data, relative frequency of responding "same" over stimulus location disparity. **(C)** BV data, mean estimation bias over stimulus location disparity. **(D)** BA data, mean estimation bias over stimulus location disparity. The subfigure-stratification and trial-binning processes for each task is identical to its group-level figure counterpart, but now with points denoting human data and lines denoting model predictions.

parameters used for the prior and noise shapes (white bar in Fig 9), the lifted SemiParametric model performs worse than our parametric models, resolving the apparent mismatch.

In conclusion, we found that our distilled parametric approach is able to explain well our data in all tasks, from unisensory localization to bisensory causal inference and localization. The proposed "distilled" parametric approach performs comparably, and possibly better depending on how complexity is measured, than very flexible semiparametric fits. After all these analyses, we can say that our participants are best described by having a mixture prior – a mix of a broad, smooth distribution (Gaussian) and a narrow, peaked distribution (Laplace) – together with a non-constant noise shape that dips strongly at the center and rises towards the periphery, in an approximately exponential fashion (Exp noise model). While our best model provides a good account of human responses in multiple tasks, there are still systematic deviations between data and model that warrant further investigation.

## Discussion

In this paper, we have systematically tested the ability of various families of Bayesian observer models in explaining both multisensory perception and causal inference. Using our dataset containing both unisensory and bisensory stimulus location estimates tasks as well as a bisensory category judgment task, we have probed in detail the role of priors and sensory noise. Using a flexible, data-driven modeling approach, our study revealed the role of non-Gaussian priors over stimulus location and eccentricity-dependent sensory noise for an improved characterization of human multisensory perception.

### Modeling analysis

We first considered the standard, "vanilla" Bayesian observer modeling assumptions common in the multisensory perception literature, which amount to spatially-constant sensory noise functions and a simple Gaussian over spatial locations. This baseline model fails to capture key features of our participant's responses in the unisensory estimation tasks, such as the strong central tendency in UV data and the lack of such tendency in UA data. We attempted to explain these discrepancies via an auditory range recalibration effect, discussed more in detail below. While improving the fits, the inclusion of recalibration in the model did not fully resolve the mismatch with the data. We hence concluded that the vanilla Bayesian observer model is an unsatisfactory description of human multisensory perception, justifying our consideration for more complex sensory noise and prior functions.

To explore potential families of eccentricity-dependent sensory noise functions and non-Gaussian priors, we developed a semiparametric modeling approach, which can flexibly represent any prior or noise shape under the assumptions of smoothness, monotonicity and central symmetry, imposed to prevent overfitting. The resulting semiparametric models explain our unisensory data remarkably well both qualitatively in terms of visual match to the data and via quantitative model comparison, hinting at the validity of these shapes in explaining unisensory perception. If such inferred shapes described objective properties of the participants' sensory systems and were not merely an exercise in curve-fitting unisensory data, we expected that the same sensory noise and prior functions would generalize to *bisensory* localization and causal inference tasks, on a participant-specific level.

Following this idea, we then "lifted" the semiparametric fits to fit the full data including bisensory tasks. Admittedly, these are not pure predictions in that we still need to fit some additional parameters, but nonetheless we prevent the models from having excessive flexibility by keeping the semiparametric shapes almost entirely frozen, and with a total number of free parameters close to that of the original vanilla model. These lifted-semiparametric model jointly explain the full data (unisensory and bisensory tasks) remarkably well and overwhelmingly better than the vanilla models, confirming our findings about the shapes of priors and sensory noise. Overall, our results highlight the effectiveness of this semiparametric approach in discovering latent properties underlying human perception, therefore demonstrating its strong potential in empirically informing the formulation of modeling assumptions for explaining human behavior.

Finally, inspired by the semiparametric models, we developed several parametric models and fitted them to either only the unisensory data or both unisensory and bisensory data together. The goal here was 'distillation', that is to provide a simpler, *parametric* description of our findings which would perform similarly well. The best-performing parametric model features an (inverted) exponential sensory noise shape and a Gaussian-Laplace mixture prior, which we discuss below. Crucially, our best parametric description fits the data qualitatively and quantitatively similarly to the full (and much more cumbersome) semiparametric model, meaning that our results can be relatively easily implemented in future modeling endeavors. We emphasize that our theoretical commitment is to the qualitative features of these models—eccentricity-dependent sensory noise and a prior with structure beyond a single Gaussian—rather than their exact functional forms. The specific parametric shapes (exponential noise, Gaussian-Laplace mixture) were chosen for simplicity and tractability, not because we believe perception literally implements these functions. Indeed, multiple parametric families (Exp-GaussianLaplace, Exp-TwoGaussians, Const-GaussianLaplace) perform comparably, suggesting that the data constrain the qualitative shape without uniquely determining the functional form.

Recent work has proposed process-level models for audiovisual integration based on correlation detection between sensory streams, which can account for causal inference phenomena with few free parameters [88]. Such algorithmic approaches address a complementary question to ours: whereas we characterize what statistical structure best describes human behavior, process-level models address how such inferences might be implemented. Bridging these levels of analysis—by identifying what architectural features of process-level models give rise to behavioral signatures like central response biases and eccentricity-dependent variability—represents a promising direction for future work.

## Multisensory and causal inference modeling results

Our analysis brings several new findings of interest for the computational study of multisensory perception and causal inference.

The semiparametric models as well as the best-performing parametric model feature a sensory noise shape with a strong dip in the center and a saturating noise level towards the periphery. Conversely, the prior distribution has a strong peak in the center and smooth, broad tails. Notably, both assumptions deviate strongly from the assumptions of homoskedastic (constant) sensory noise and Gaussian priors commonly adopted in previous computational work.

In contrast to quadratic sensory noise shapes assumed in previous models [13], the success of our exponential sensory noise function suggests a saturation of noise levels towards peripheral eccentricity angles even for relatively narrow stimulus ranges. Our results are qualitatively compatible with sinusoidal sensory noise shapes [29,30], which when restricted to a narrow range may also look like saturating curves depending on the chosen periodicity. The large mass of the Gaussian-Laplace mixture prior in central angles is consistent with past literature [47,84], reflecting particularly strong human expectations for stimuli to appear in the center-front. This prior expectation appears even stronger than what the current models could capture, as the Semiparametric and Gaussian-Laplace models were unable to reproduce the strong UV central tendencies in the high visual reliability condition, when the stimulus location is in in [–8.6, 2.9]° or [2.9, 8.6]° (Figs 5, 8, and Fig N in S1 Appendix Section C). We believe this was not due to trivial experimental design choices—for example, the cursor position was not reset to zero in each trial, preventing the formation of a central tendency effect just out of "laziness" in moving the cursor away from zero [89]. As our results suggest, incorporating these sensory noise and prior assumptions is crucial for developing models that faithfully reflect human multisensory perception.

In bisensory perception, the causal inference strategy—how people combine estimates derived from same and different source assumptions respectively—is a key part of Bayesian observer models. As our model comparison results suggest, the Model Averaging (MA) and Probability Matching (PM) strategies describe human responses better than Model Selection (MS), indicating a more nuanced strategy. The superiority of probability matching is consistent with past works in perception [25,40], while the indistinguishability between PM and MA in our study necessitates further investigation of the task conditions that elicit either strategy in human observers.

A side finding of our modeling endeavor is that for bisensory tasks, our models included multiplicative scaling parameters for the visual and auditory sensory noise with respect to unisensory noise. A multiplicative factor of $1\times$ would mean that visual and auditory noise in the bisensory trials is the same as the sensory noise measured in the unisensory trials alone, which is the common assumption. Instead, the model fits consistently found that the noise level was *higher*—by a factor of $1.1\times$ to $1.6\times$ compared to unisensory perception—during bisensory perception. We hypothesize this increased noise may be due to attention diversion caused by collecting observations from different sensory modalities. Still, this finding is noteworthy since a basic assumption of much of the multisensory modeling literature is that multisensory perception can be predicted by unisensory perception alone [2,4,5], but here there seems to be an additional interaction which deserves further investigation. Recent causal inference studies have considered increases in sensory noise under bimodal conditions [69,90,91], although most prior work in multisensory causal inference did not use unisensory data to inform multisensory modeling [1,18].

A final experimental finding from our data is the presence of auditory range recalibration. We believe this effect arises from some aspects of our experimental design, specifically from the different visual and auditory stimulus ranges used in our experiment. Using model comparison, we found that Bayesian observer models that recalibrate (stretch) noisy auditory observations to the visual range describe our participants' data better compared to models that do not. Validating our modeling assumptions, the magnitude of the recalibration parameter we find matches very closely the ratio between true stimulus ranges of the two sensory modalities (4/3). While this effect might be unique and incidental to our experimental setup, it is worth noting and may deserve a separate investigation at the intersection of multisensory perception, causal inference, and recalibration.

## Limitations and future directions

Our study has several limitations that suggest directions for future research.

First, because our experimental design focused on specific spatial ranges and sensory modalities, extending the stimulus range and incorporating other modalities could help determine whether our findings generalize across different contexts. While our modeling framework is general, our findings may partly reflect features of our experimental setup. Other audiovisual causal inference datasets [92,93], could provide valuable tests of whether the qualitative features we report generalize across paradigms and applying our modeling techniques to other datasets is a promising direction for future work.

Second, we acknowledge that the auditory range recalibration effect—where participants appeared to stretch the auditory stimulus range to match the visual range—may be specific to our experimental setup with mismatching visual and auditory ranges. Additionally, our use of a visual pointer for all responses means that auditory-only trials required cross-modal responding, which could in principle influence behavior; we suspect this may contribute to the recalibration effect, though this is unlikely to affect our core findings, which derive from the structure of unisensory response distributions rather than cross-modal comparisons. Still, our findings hint more generally at an interplay between sensory recalibration and multisensory causal inference which deserves further investigation [69]. For unisensory conditions, the recalibration effect could alternatively be explained by assuming different location priors for vision and audition, with the auditory prior being relatively uninformative and thus producing minimal empirical bias in posterior location estimates. However, extending this modality-specific-prior account to bisensory trials would require a nontrivial modification of the model class, because both visual and auditory inputs may originate from a common source—which necessitates a shared prior $p(s)$ for evaluating the posterior probability of the $C=1$ hypothesis. In other words, relaxing the supramodal prior assumption would require an entirely new class of models for Bayesian causal inference, which represents an interesting direction for future work. Notably, although the evidence remains circumstantial, the fitted recalibration parameters being approximately 4/3—which corresponds to the ratio of visual to auditory stimulus ranges in our experiment—lends support to our interpretation, as this correspondence would otherwise be difficult to explain. In line with past work [49,66–74], we propose that the recalibration mechanism may extend to multisensory causal inference and encourage future research to test its generality.

Third, our semiparametric modeling approach, though flexible in capturing complex sensory noise and prior shapes, involves a high number of free parameters which makes it a challenging and expensive model fitting problem. We mitigated the risk of overfitting by comparing to distilled parametric models with fewer parameters and using multiple model selection criteria. Still, future research could attempt to employ efficient Bayesian inference methods [94] and other techniques to extend the validation of our models and make fitting more efficient.

Fourth, we assumed static and symmetric priors and sensory noise functions centered at zero, but in real-world contexts, priors may be dynamic [95,96] and sensory noise may exhibit asymmetries due to attentional biases or contextual factors [97–99]. In parallel to studies in higher-level cognition suggesting the flexibility of human priors for sophisticated events [100], it could be possible that similar top-down mechanisms are at play during human causal inference following initial perception [101]. Future work could clarify the environmental contexts that elicit specific prior assumptions, and investigate the optimality of such adjustments given potentially non-Gaussian statistics in the real world (especially in the absence of feedback as in the current experiment). Overall, incorporating adaptive priors and asymmetric noise functions could provide a more accurate representation of human perception.

Fifth, although we varied the sensory noise and prior components in a semiparametric manner, the causal inference strategies used in the bisensory conditions could not be encoded as flexibly. Instead, we adopted several well-established strategies from the literature, including model selection, model averaging, and probability matching. Recent work has suggested that human causal inference may also rely on heuristic or rule-based mechanisms [13,69,91,102–104], which we did not systematically examine here. Future research could aim to unify these diverse causal inference strategies within a semiparametric framework, which would enable a more systematic comparison of their descriptive power.

Finally, the increased sensory noise observed in bisensory conditions suggests complex interactions between sensory modalities involving attentional or cognitive factors not fully accounted for in our models [105]. Future research should explore the neural and cognitive mechanisms underlying this increased noise, perhaps through neuroimaging or attentional manipulations [106], to disentangle sensory and cognitive contributions.

## Conclusions

Our study reveals stimulus-dependent (eccentricity-dependent) sensory noise and complex (non-Gaussian) priors as critical determinants of multisensory perception. Using a semiparametric approach, we found that sensory noise decreases centrally and plateaus peripherally, while the prior distribution shows a narrow central peak with broad tails—patterns that deviate substantially from conventional assumptions. We distilled these findings into practical parametric models using exponential sensory noise and Gaussian-Laplace mixture priors, providing more accurate yet parsimonious tools for modeling human perception. We expect similar parametric families to be applicable to other studies. Moreover, the evidence for auditory range recalibration and increased bisensory noise in our detailed analysis suggests complex adaptive mechanisms in sensory integration. By refining these longstanding modeling assumptions, our work provides new tools for multisensory perception research and for future investigations into the neural and cognitive mechanisms underlying sensory integration and causal inference.

## Methods

### Ethics statement

The study protocol was approved by the Institutional Review Board of Baylor College of Medicine (protocol number H-23976), and all participants provided informed written consent.

### Participants

Fifteen participants participated (12 women; between 18 and 30 years of age). All had normal or corrected-to-normal vision and none had known hearing impairments. Participants received $60 for their participation.

## Apparatus and stimuli

Auditory stimuli were presented using an array of 7 equally spaced speakers (at center-to-center separation of 6 cm), mounted in a horizontal row on a custom-built wooden frame. The frame was covered with a black Lycra cloth that acted as the projector screen and also concealed the locations of the speakers. Visual stimuli were projected using a ceiling-mounted projector with a resolution of 1024 by 768 pixels at 60 Hz. A Harmony Audio multi-channel sound device controlled all 7 speakers via FireWire. The visual and auditory stimuli were controlled through Matlab on a MacBook.

The auditory stimulus was a 25 ms sinusoidal wave with a frequency that was selected randomly from a uniform distribution between 900 and 1100 Hz, to reduce the possibility that the participant could identify individual speakers by their characteristic sound at a particular frequency.

Each visual stimulus was a spot with a two-dimensional, radially symmetric Gaussian luminance profile on a dark background. On each trial, the radius (standard deviation) of the Gaussian spot was randomly chosen from three possible values: 21, 130, and 250 pixels, which at a seating distance of 60 cm corresponds to 1.9 (high reliability level), 11.6 (medium reliability level), and 22.3 degrees (low reliability level) of visual angle. We consider trials with the same visual stimuli radius as belonging to the same visual reliability level, leading to a total of 3 possible visual reliability levels. The visual stimulus was displayed for 16.7 ms (1 frame). A white fixation cross was continuously present in the center of the screen (at 507 pixels from the left edge).

Participants were seated on a height-adjustable stool at a distance of approximately 60 cm from the projection screen, so that the angular separation between speaker centers was about 5 degrees and the speaker array was at eye level. Anechoic foam was applied to the walls of the room to reduce echo.

Given this setup, we can convert pixels on the screen to degrees (°) of visual or auditory angle, which would be used as the units for our subsequent analysis. There are 0.08928 degrees per pixel. Hence, the response range is $[-45°, 45°]$, the visual stimulus range is $[-20°, 20°]$, and the auditory stimulus values are always in $\{0°, \pm 5°, \pm 10°, \pm 15°\}$. Note that there is a disparity between the wider visual stimulus range and narrower auditory stimulus range. This has motivated us to include an auditory range recalibration parameter $\rho_A$ for fitting auditory data in the models later on. Also due to the narrow stimulus and response ranges, we have used priors and sensory noise functions defined on the continuous real numbers, instead of those defined on a circle.

## Experimental procedure

Trials were organized in four types of blocks: *unisensory auditory localization* (UA), *unisensory visual localization* (UA), *bisensory localization* (B), and *bisensory causal judgment* (BC, where $C = 1$ corresponds to "same" and $C = 2$ correspond to "different"). The bisensory localization blocks contained interleaved trials of two response types (BV, BA), so there were five task types in total.

In UV and UA blocks, the instructions given to participants were: "*Put your head in the chin rest to ensure your head is always in the same spot. Then look straight ahead and there will be a brief beep or a flash. Then use the mouse on the table to move the vertical line to the spot where you think it happened, and then click.*" A trial was structured as follows. A screen only showing the fixation cross appeared for 1 second. Then, the stimulus was presented, followed by a 200ms screen showing only the fixation cross. After that, the participant used the mouse to move a vertical cursor, 1 pixel wide and locked to the horizontal midline, to the perceived location of the stimulus and clicked to submit the response. The cursor was not re-initialized between trials; at trial onset, it remained at the horizontal position where it had been left by the participant's click on the previous trial. In UV blocks, the radius of the visual stimulus was randomly drawn from the three possible radii and its center was located on the horizontal midline at a distance drawn from a uniform distribution between 283 and 731 pixels from the left edge (corresponding to $[-20°, 20°]$ degrees). In UA blocks, the auditory stimulus was presented through one of the 7 speakers (located at $\{0°, \pm 5°, \pm 10°, \pm 15°\}$ degrees), chosen randomly with equal probability.

In B blocks, the visual and auditory stimuli were presented simultaneously. The auditory stimulus was again presented through a randomly chosen speaker. With a 50% probability, the visual stimulus was shown in the same location as that speaker. Otherwise, its location was drawn in the same way as in the UV blocks (independently from the auditory stimulus). Participants were instructed: "*There will be a flash and a beep at the same time. Use the mouse to indicate the location of the flash OR the beep, whichever the screen says at the beginning. Sometimes they will be in the same location and sometimes they won't.*" They were not given any information on the 50% same-location probability. On each trial, the participant had to report the location of either the visual or the auditory stimulus, randomly with equal probability. The participant was informed of the required response type (BA or BV) through a screen instruction shown during the response period.

In BC blocks, the stimuli were generated in the same way as in the bisensory localization blocks. Participants were instructed: "*There will be a flash and a beep at the same time. Your job is to judge if you think they were in the same location or not.*" As above, they were not informed about the 50% same-location probability. The participant judged whether or not the auditory and the visual stimulus were presented at the same location by clicking on a "yes" or a "no" button presented on the screen. The experiment consisted of 5 sessions on separate days. Each session consisted of 6 blocks: 1 UV, 1 UA, 2 B, and 2 BC. During the first session, the order of the blocks was (UV, UA, B, BC, B, BC). On sessions 2–5, the order of the blocks was randomized. Each block contained 100 trials. In total, each participant completed 500 UV trials, 500 UA trials, 1000 B trials, and 1000 BC trials, for a total of 3000 trials. participants were advised to take a short break between blocks.

Each participant received verbal and on-screen instructions at the start of the first session. participants then performed 40 practice trials, 10 of each block type. To ensure understanding of the task, the on-screen instructions were redisplayed before each block type began.

Due to the dual-monitor setup in the experiment room, some participants accidentally moved the cursor off their response monitor on to the other monitor before clicking the mouse. Trials in which the responses were out of the response range were discarded (this involves discarding 18 trials across all participants, which is 0.0404% of the 44600 trials collected).

## Bayesian observer models for auditory-visual localization and causal inference

In this paper, we use a family of Bayesian observer models to fit human data collected on the five task types. Bayesian observer models provide a normative framework for causal inference and cue integration—-in other words, they reflect the optimal strategy for such tasks, amid various sources of noise that corrupt our sensory observations [2]. Bayesian observer models assume a particular external generation process, which specifies how noise sensory observations $x$ are generated from the true underlying structure of the world $s$. Given that the decision-maker only has access to such noisy observations, Bayesian observer models assume that decision-makers internally infer the true underlying structure from the noisy observations via Bayes' rule: $p(s|x) \propto p(x|s)p(s)$ [10]. This formula specifies how the likelihood function of the underlying true parameter $p(x|s)$ and the decision-maker's prior distribution $p(s)$, reflecting subjective knowledge about the distribution of $s$ before receiving sensory evidence, give rise to a posterior distribution $p(s|x)$, which guides the estimation of $s$ conditioned on the participant's noisy observation $x$. Note that in Bayes' rule the likelihood $p(x|s)$ is a function of $s$ for a fixed measurement $x$; when the same expression is rewritten as a function of $x$, it becomes the measurement distribution, which reflects the generation process of $x$ from $s$. Given the ecological relevance of cue integration and causal inference, it is reasonable to assume that evolutionary pressure has forced humans to adopt near-Bayes-optimal solutions to these tasks. These considerations have motivated the usage of Bayesian observer models in perceptual causal inference and in our paper [1,2,4–6,14,18,24–26].

## Bayesian modeling procedures

**UV and UA tasks.** In UV and UA tasks, for a particular trial, as experimenters we have access to the true stimulus location $s$ and the participant-reported stimulus location estimate $r_s$. In this paper, we denote with $\mathcal{N}(\mu, \sigma^2)$ a normal distribution with mean $\mu$ and variance $\sigma^2$.

As the first modeling step, we define the generative process. We assume participants do not have direct access to $s$, but instead receive a noisy measurement $x|s \sim \mathcal{N}(x; s, \sigma^2(s))$, where $\sigma(s; \boldsymbol{\theta}_\sigma)$ is an eccentricity-dependent sensory noise function with free parameters $\boldsymbol{\theta}_\sigma$ (omitted later). Separate sensory functions $\sigma_V(s), \sigma_A(s)$ are defined for vision and audition, and used in visual and auditory trials respectively. Their corresponding parameters $\boldsymbol{\theta}_\sigma$ are independently defined and denoted $\boldsymbol{\theta}_V$ and $\boldsymbol{\theta}_A$ respectively.

Thus, we define the measurement distribution $p(x|s) = \mathcal{N}(s, \sigma^2(s))$. For vision, there are three reliability levels. Hence, we have different sensory noise functions $(\sigma_V(s), \alpha_{\mathrm{med}} \cdot \sigma_V(s), \alpha_{\mathrm{low}} \cdot \sigma_V(s))$ for each level, where $(\alpha_{\mathrm{med}}, \alpha_{\mathrm{low}})$ are coefficients applied to the medium- and low-reliability trials, respectively, but $\boldsymbol{\theta}_\sigma$ remains the same within each participant.

As the second modeling step, we define the inference process: we assume that the participant possesses a prior over the true stimulus location $p(s; \boldsymbol{\theta}_{\mathrm{prior}})$ that is centered at $s = 0$, and parameterized by $\boldsymbol{\theta}_{\mathrm{prior}}$ (omitted later). Hence, the participant infers the stimulus location from the noisy measurement $x$ using Bayes' rule: $p(s|x) \propto p(x|s)p(s)$, where all three functions involved are functions of $s$.

As the final modeling step, we assume the participant reports their posterior mean estimate $\hat{s} = \int_{-\infty}^{\infty} s \cdot p(s|x) ds$, corrupted by motor noise. Hence the final reported location estimate is generated by $r \sim \mathcal{N}(\hat{s}, \sigma^2_{\mathrm{motor}})$, in which $\sigma_{\mathrm{motor}}$ is the standard deviation of the motor noise. We also account for lapse trials, in which the response is uniformly sampled from the response range $r \sim U(-45, 45)$. Each trial has an independent probability $\lambda$ (lapse rate) of being a lapse trial. For an exploration of alternative lapse distribution families, see S1 Appendix Section C.

In many of our models, we also introduced an auditory range recalibration parameter $\rho_A$ which rescales the noisy observation $x_A$ before computing the likelihood, effectively stretching the auditory range (see later for details). The $\rho_A$ parameter is relevant for all tasks with auditory stimuli, so all but UV trials.

**BC task.** In the BC task type, for a particular trial, we have the true stimulus locations $(s_V, s_A)$ and the participant-reported causal judgment $r_C \in \{1, 2\}$ (where 1 stands for 'same' location). We can obtain the posterior probability of each category judgment by Bayes' rule: $p(C|x_V, x_A) \propto p(x_V, x_A|C)p(C)$. The prior for category judgment $p_{\mathrm{same}} = p(C = 1)$ is a free model parameter. We can obtain the likelihood $p(x_V, x_A|C)$ as a function of $(x_V, x_A)$ for each possible value of $C \in \{1, 2\}$. For $C = 1$ (same), we assume that $x_V, x_A \overset{\mathrm{iid}}{\sim} \mathcal{N}(s, \sigma^2(s))$, so that, assuming a shared supra-modal prior $p(s) \equiv p(s_V) = p(s_A)$:

$$p(x_V, x_A|C = 1) = \int_{-\infty}^{\infty} p(x_V, x_A|s)p(s)ds = \int_{-\infty}^{\infty} p(x_A|s)p(x_V|s)p(s)ds,$$

(1)

For $C = 2$ (different), we assume independent $s_V, s_A$:

$$p(x_V, x_A|C = 2) = \int_{-\infty}^{\infty} p(x_A|s_A)p(x_V|s_V)p(s_V, s_A)ds_A ds_V$$

$$= \int_{-\infty}^{\infty} p(x_A|s_A)p(s_A)ds_A \int_{-\infty}^{\infty} p(x_V|s_V)p(s_V)ds_V,$$

(2)

We can then obtain posterior probabilities $p(C|x_V, x_A)$ for $C \in \{1, 2\}$ via Bayes' rule, and use some causal inference strategy $g(\cdot)$ to map them to an estimate $\hat{C}$ (to be explained later), which is also the response $r_C$ in non-lapse trials:

$$\hat{C} = g(p(C = 1|x_V, x_A), p(C = 2|x_V, x_A)).$$

(3)

For lapse trials generated with probability $\lambda$, we have the response $r_C$ sampled randomly and uniformly in $\{1, 2\}$.

**BV and BA tasks.** For BV and BA trials with true stimulus locations $(s_V, s_A)$ and response $r_A$ (for BA trials only) or $r_V$ (for BV trials only), we need a couple more steps beyond the BC model. After we obtain posterior probabilities $p(C|x_V, x_A)$

for $C \in \{1, 2\}$ identically to BC trials, we need to compute the posterior mean estimate of the stimulus location ($\hat{s}_A$ for BA, $\hat{s}_V$ for BV) under both hypotheses $C = 1$ and $C = 2$.

Without loss of generality, assume that we are in a BA trial. Hence:

$$\hat{s}_{A,C=1} = \int_{-\infty}^{\infty} s \cdot p(s|x_V, x_A)ds = \int_{-\infty}^{\infty} s \cdot p(x_V, x_A|s)p(s)ds$$
$$= \int_{-\infty}^{\infty} s \cdot p(x_A|s)p(x_V|s)p(s)ds \tag{4}$$

$$\hat{s}_{A,C=2} = \int_{-\infty}^{\infty} s_A \cdot p(s_A|x_A)ds_A = \int_{-\infty}^{\infty} s_A \cdot p(x_A|s_A)p(s_A)ds_A \tag{5}$$

We make the additional assumption that for our participants, the prior distributions for visual and auditory stimulus locations are identical: the possibility of participants committing to the hypothesis $s_V = s_A$ implies $p(s_V) = p(s_A)$. Under this assumption, we can then use some causal inference strategy and motor noise (collectively denoted by stochastic function $g(\cdot)$; to be explained later) to generate a response $r_A$:

$$r_A \sim g(\hat{s}_{A,C=1}, p(C = 1|x_V, x_A), \hat{s}_{A,C=2}, p(C = 2|x_V, x_A), \sigma_{\text{motor}}) \tag{6}$$

For lapse trials generated with probability $\lambda$, we have $r_A \sim U(-45, 45)$.

For BV trials, we just need to compute $\hat{s}_{V,C=1}$ identically to $\hat{s}_{A,C=1}$, and:

$$\hat{s}_{V,C=2} = \int_{-\infty}^{\infty} s_V p(s_V|x_V)ds_V = \int_{-\infty}^{\infty} s_V p(x_V|s_V)p(s_V)ds_V \tag{7}$$

Then, we follow a similar procedure to obtain the response $r_V$.

### Sensory noise and prior functions

The above model formulations depend on several expressions that have not been defined yet. For UV and UA tasks, this includes the prior over stimulus locations $p(s)$ and the sensory noise functions $\sigma_V(s)$ and $\sigma_A(s)$, respectively. For BC, BV, and BA tasks, we additionally need to define the causal inference strategy $g(\cdot)$. Below, we first introduce our semiparametric definitions for prior and sensory noise shapes, used to reveal the underlying prior and sensory noise functions shapes from human data (see Fig 4). Subsequently, we define the prior and sensory noise parametric families inspired by the semiparametric fits.

**Semiparametric prior and sensory noise shapes.** For the semiparametric fits, the prior and sensory noise function were defined based on their values at 12 "pivot" points along the stimulus range: $s_j \in \{0, 0.1, 0.3, 1, 2, 4, 6, 8, 10, 15, 20, 45\}$ for $0 \leq j < 12$. We assume that these functions are symmetric with respect to $s = 0$.

To ensure that $\sigma_V(s), \sigma_A(s)$ are positive and monotonically increasing as a function of $|s|$, we encode $\sigma(s_0)$ and the differences between adjacent pivot values, $\Delta\sigma(s_j) = \sigma(s_j) - \sigma(s_{j-1})$, as log-values. Taking the exponential ensures positivity of each term, and a cumulative sum operation allows us to compute $\sigma(s_j)$ at any pivot $j$. Due to the assumed central symmetry, we mirror along 0 to obtain values at negative pivot points. Finally, cubic interpolation is used to generate smooth function values between pivot points.

For vision, there are three reliability levels. Hence, we have different sensory noise functions $(\sigma_V(s), \alpha_{\text{med}} \cdot \sigma_V(s), \alpha_{\text{low}} \cdot \sigma_V(s))$ for each level, where $(\alpha_{\text{med}}, \alpha_{\text{low}})$ are coefficients applied to the medium- and low-reliability trials, respectively.

The semiparametric prior $p(s)$ is built in a similar manner, defining values of $\log p(s)$ at the pivots ensuring that the (log) prior is monotonically decreasing as a function of $|s|$. Interpolation is performed in log space, followed by exponentiation and normalization via numerical integration to obtain a valid probability density function over the full range. Notably, due to the normalization, there is one fewer degree of freedom, so we fix $\log p(s_0) = 0$ (any choice is equivalent, as we normalize the pdf at the end).

**Parametric prior families.** We tested the following parametric priors over stimulus locations:

• A Gaussian prior (SingleGaussian):

$$p(s) = \mathcal{N}(s; 0, \sigma_s^2).$$

(8)

• A two-Gaussian mixture prior (TwoGaussians):

$$p(s) = (1 - \omega) \cdot \mathcal{N}(s; 0, \sigma_s^2) + \omega \cdot \mathcal{N}(s; 0, (\sigma_s + \sigma_\Delta)^2).$$

(9)

• A Gaussian-Laplace mixture prior (GaussianLaplace):

$$p(s) = (1 - \omega) \cdot \mathcal{N}(s; 0, \sigma_s^2) + \frac{\omega}{2b} \exp\left(-\frac{|s - 0|}{b}\right).$$

(10)

These three prior families are increasingly concentrated at the center $s = 0$. The free parameters for the prior are hence $\boldsymbol{\theta}_{\text{prior}} = \sigma_s$ for the SingleGaussian prior, $\boldsymbol{\theta}_{\text{prior}} = (\sigma_s, \sigma_\Delta, \omega)$ for the TwoGaussians mixture prior, and $\boldsymbol{\theta}_{\text{prior}} = (\sigma_s, b, \omega)$ for the Gaussian-Laplace mixture prior.

**Parametric sensory noise families.** Regarding the eccentricity-dependent sensory noise, we have:

• The homoskedastic (Const) noise:

$$\sigma(s) = \sigma_0.$$

(11)

• Heteroskedastic exponential (Exp) sensory noise, which dips at the center $s = 0$ and plateaus towards the periphery:

$$\sigma(s) = \sigma_0 + k_1(1 - \exp(-k_2|s|)).$$

(12)

The free parameters for sensory noise are $\boldsymbol{\theta}_\sigma = \sigma_0$ for the homoskedastic noise and $\boldsymbol{\theta}_\sigma = (\sigma_0, k_1, k_2)$ for the heteroskedastic exponential noise. We assume that the sensory noise functions for vision $\sigma_V(s)$ and audition $\sigma_A(s)$ belong to the same function family (Const or Exp), but their parameters $\boldsymbol{\theta}_V, \boldsymbol{\theta}_A$ may differ.

For vision, there are three reliability levels. Hence, we have different sensory noise functions $(\sigma_V(s), \alpha_{\text{med}} \cdot \sigma_V(s), \alpha_{\text{low}} \cdot \sigma_V(s))$ for each level, where $(\alpha_{\text{med}}, \alpha_{\text{low}})$ are coefficients applied to the medium- and low-reliability trials, respectively, but $\boldsymbol{\theta}_\sigma$ remains the same within each participant. Note that $(\alpha_{\text{med}}, \alpha_{\text{low}})$ are also free parameters of the model.

Examples of the parametric prior and sensory noise function families are visualized in Fig 13.

## Causal inference strategy for bisensory modeling

In our models, we considered three alternative causal inference strategies: Model Selection (MS) [1,23,25], Model Averaging (MA) [1,3,25], and Probability Matching (PM) [25]. Relevant to only the bisensory tasks BC, BV, and BA, each causal

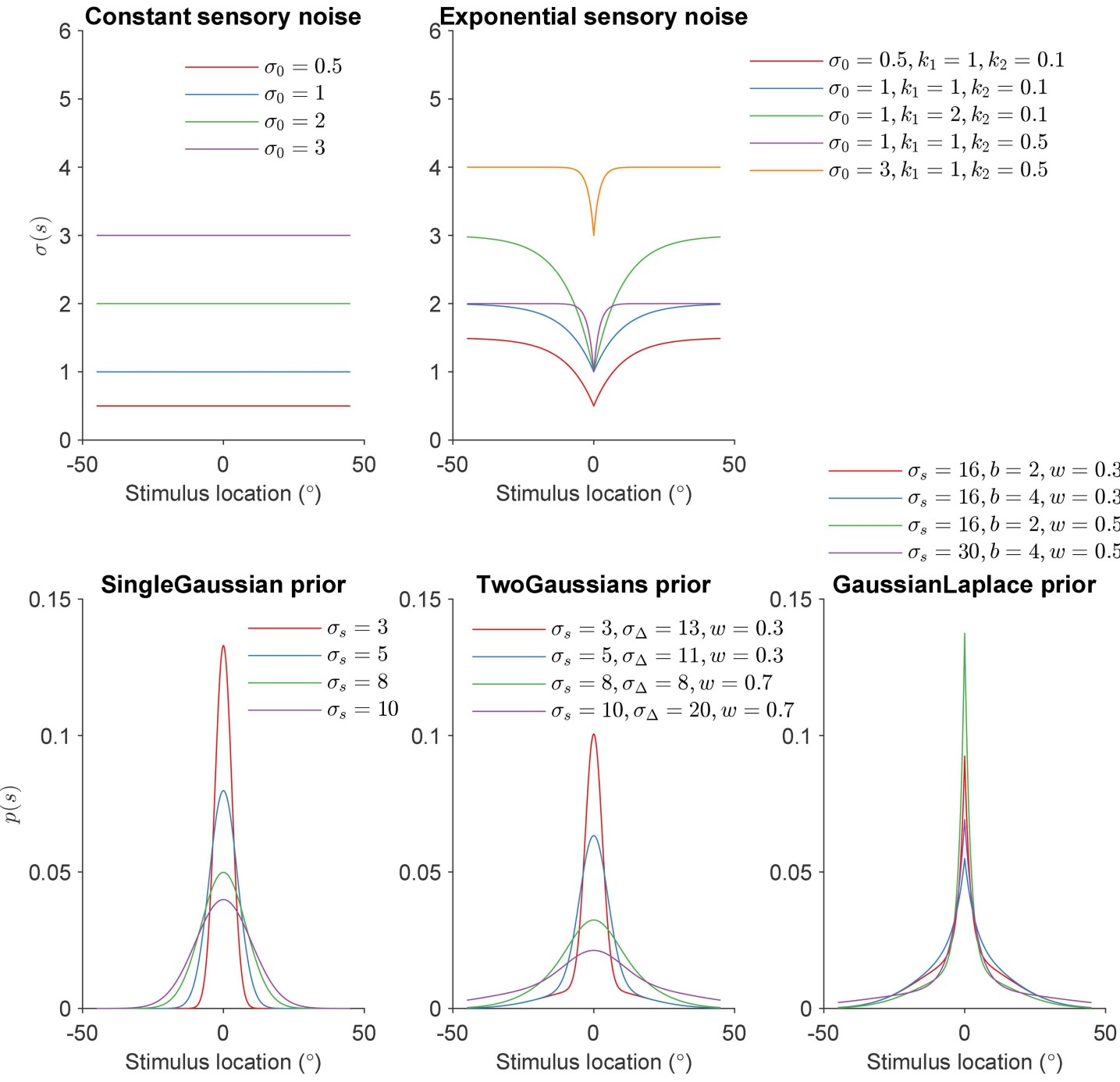

**Fig 13. Illustrative examples of the sensory noise families and prior parametric families used in our parametric Bayesian observer models.** All shapes are defined over the range [–45°, 45°].

inference strategy is a (possibly stochastic) mapping $g(\cdot)$ from the two posterior distributions under each hypothesis $C = 1$, $C = 2$ to a response for the task type at hand.

- For BC trials, in both MS and MA strategies (which are equivalent for BC trials), upon observing noisy measurements $x_V$ and $x_A$, the observer will select the scenario $\hat{C}$ with highest posterior probability, $p(C = \hat{C}|x_V, x_A) > 0.5$. To compute the empirical response probability, we integrate over the latent noisy observations according to the generative model,

$$p(\hat{C}|x_V, x_A) = \int_{-\infty}^{\infty} \mathbb{1}_{p(C=\hat{C}|x_V, x_A) > 0.5} \cdot p(x_A|s_A)p(x_V|s_V)dx_A dx_V. \tag{13}$$

- In contrast, in the PM strategy, we integrate over all possible values of $x_V, x_A$ weighted by the posterior value $p(C = \hat{C}|x_V, x_A)$, instead of skipping $x_V, x_A$ values that lead to $p(C = \hat{C}|x_V, x_A) \leq 0.5$.

$$p(\hat{C}|x_V, x_A) = \int_{-\infty}^{\infty} p(C = \hat{C}|x_V, x_A) \cdot p(x_A|s_A)p(x_V|s_V)dx_A dx_V. \tag{14}$$

The final response probability for both causal inference strategies additionally includes a probability of lapsing, which is a free parameter $\lambda$. For BC trials, lapses are defined as randomly selecting each response with equal probability:

$$p_{\text{final}}(\hat{C}|x_V, x_A) = \frac{\lambda}{2} + (1 - \lambda) \cdot p(\hat{C}|x_V, x_A). \tag{15}$$

For BV and BA trials, the three causal inference strategies all operate on the posterior stimulus location estimates reached under each hypothesis (same/different source judgment; a.k.a., $C = 1$ or $2$), denoted as $\hat{s}_{C=1}, \hat{s}_{C=2}$ respectively.

- In MS, the final stimulus location estimate $r$ is the posterior estimate among $\hat{s}_{C=1}, \hat{s}_{C=2}$ that is associated with the greater posterior probability $p(C = i|x_V, x_A)$, corrupted by Gaussian motor noise:

$$p(r|x_V, x_A) = \mathcal{N}\left(r; \hat{s}_{C=\text{argmax}_i\{p(C=i|x_V, x_A)\}}, \sigma_{\text{motor}}^2\right). \tag{16}$$

- In MA, the final estimate $r$ is a weighted sum of the two posterior estimates, where the weights are the respective posterior probabilities $p(C = i|x_V, x_A)$, corrupted by Gaussian motor noise:

$$p(r|x_V, x_A) = \mathcal{N}\left(r; \sum_{i=1}^{2} \hat{s}_{C=i} \cdot p(C = i|x_V, x_A), \sigma_{\text{motor}}^2\right). \tag{17}$$

- In PM, the observer reports one out of the two posterior estimates (corrupted by Gaussian motor noise) probabilistically based on their corresponding posterior probabilities:

$$p(r|x_V, x_A) = \sum_{i=1}^{2} p(C = i|x_V, x_A) \cdot \mathcal{N}\left(r; \hat{s}_{C=i}, \sigma_{\text{motor}}^2\right). \tag{18}$$

Finally, in all models a lapse component is added to the response probability. The probability of lapsing is a free parameter $\lambda$. For BV and BA trials, lapses are defined as providing a uniformly random response within the response range $[-45°, 45°]$:

$$p_{\text{final}}(r|x_V, x_A) = \frac{\lambda}{45° - (-45°)} + (1 - \lambda) \cdot p(r|x_V, x_A).$$

(19)

## Auditory range recalibration

We introduced an auditory range recalibration parameter $\rho_A$ to account for the fact that the visual stimulus range (from $-20°$ to $20°$) is wider than the auditory stimulus range (from $-15°$ to $15°$). This has the potential to create a "stretching" effect on the auditory noisy observations towards the visual range under the ventriloquism aftereffect [66–69].

In our models featuring auditory-visual recalibration formulation, we applied $\rho_A$ as a scalar multiple on the noisy auditory observations $x_A$ immediately after their generation from the true stimuli, before evaluating $p(x_A|s_A)$ in the model. Importantly, the effects of this recalibration is never corrected in later inference stages of the model. Note that the value $\rho_A = 1$ removes the auditory range recalibration completely.

## Models and free parameters

**Semiparametric model.** The parameters of the semiparametric model include:

- Semiparametric sensory shapes $\boldsymbol{\theta}_V$ and $\boldsymbol{\theta}_A$, defined at the 12 pivot points each; the two sensory shapes amount to 24 parameters in total.

- Semiparametric prior parameters $\boldsymbol{\theta}_{\text{prior}}$ defined at the pivot points; this amounts to 11 prior shape parameters (one pivot value is fixed due to the normalization).

- Noise scale parameters $\alpha_{\text{med}}$ and $\alpha_{\text{low}}$ for medium- and low-reliability, respectively.

- Motor noise $\sigma_{\text{motor}}$ and lapse rate $\lambda$.

- The range recalibration parameter $\rho_A$.

- The semiparametric model was only fitted to unisensory data, so there is no choice of causal inference strategy.

Overall, there are 40 free parameters per participant in the semiparametric model:

$$\boldsymbol{\theta} = \left( \boldsymbol{\theta}_V^{(1:12)}, \boldsymbol{\theta}_A^{(1:12)}, \boldsymbol{\theta}_{\text{prior}}^{(1:11)}, \alpha_{\text{med}}, \alpha_{\text{low}}, \lambda, \sigma_{\text{motor}}, \rho_A \right).$$

**Lifted-semiparametric models.** For the lifted-semiparametric fits over all tasks, the 3 causal inference strategies lead to only 3 models to be fitted.

The functions $\sigma_V(s), \sigma_A(s), p(s)$ are adopted from the semiparametric model fits and are hence fixed. However, we still have free parameters $(\alpha_{\text{med}}, \alpha_{\text{low}}, \lambda, \sigma_{\text{motor}}, \rho_A)$, which are applied in a fashion identical to the semiparametric model. Below are the additional free parameters for lifted-semiparametric models:

- The prior probability that the visual and auditory stimuli come from the same location $p_{\text{same}} = p(C = 1)$, which is used for bisensory tasks only.

- Unisensory-bisensory rescaling parameters $(\beta_V, \beta_A)$, introduced due to a potential increase in sensory noise from unisensory to bisensory trials due to attention diversion. These parameters are scalar multiples applied to the entire

sensory noise functions $\sigma_V(s), \sigma_A(s)$ respectively, increasing the function values uniformly across the stimulus range. They are only applied to bisensory tasks and not unisensory tasks.

- Prior shape parameter $\gamma$. We introduced it to allow for changes in the shape of the prior $p(s)$ during the lifted-sempiparametric fits; it is applied as an exponent to the semiparametrically fitted $p(s)$ shape and used in both unisensory and bisensory tasks.

Overall, each lifted-semiparametric model contains 9 free parameters:

$$\boldsymbol{\theta} = (\alpha_{\text{med}}, \alpha_{\text{low}}, \lambda, \sigma_{\text{motor}}, \rho_A, p_{\text{same}}, \beta_V, \beta_A, \gamma)$$

**Parametric models.** For the parametric models fitted on all data, we would need combine different priors/sensory noise/causal inference strategies. We considered 4 sensory noise and prior combinations, as well as the 3 causal inference strategies, leading to 12 unique parametric models. Using the notation described in this subsubsection below, the 4 combinations considered are Exp-GaussianLaplace, Exp-SingleGaussian, Const-GaussianLaplace, and Exp-TwoGaussians.

Each parametric model has the following free parameters:

- Parameters for the sensory noise function for vision $\sigma_V(s)$, denoted as $\boldsymbol{\theta}_V$.

- Parameters for the sensory noise function for audition $\sigma_A(s)$, denoted as $\boldsymbol{\theta}_A$.

- Parameters for the prior distribution $p(s)$, denoted as $\boldsymbol{\theta}_{\text{prior}}$.

The number of the above free parameters depends on the parametric prior distribution and sensory noise function families used by the model. Other free parameters include $(\alpha_{\text{med}}, \alpha_{\text{low}}, \lambda, \sigma_{\text{motor}}, \rho_A, p_{\text{same}}, \beta_V, \beta_A)$, which are defined and applied in a fashion identical to the lifted-semiparametric models.

Overall, across the 12 parametric models fitted on all data, the models with fewest parameters contain 11 parameters (Const-SingleGaussian-MS/MA/PM), and the models with most parameters contain 17 parameters (Exp-GaussianLaplace-MS/MA/PM, Exp-TwoGaussians-MS/MA/PM). Their parameters are:

$$\boldsymbol{\theta} = (\boldsymbol{\theta}_V, \boldsymbol{\theta}_A, \boldsymbol{\theta}_{\text{prior}}, \alpha_{\text{med}}, \alpha_{\text{low}}, \lambda, \sigma_{\text{motor}}, \rho_A, p_{\text{same}}, \beta_V, \beta_A)$$

**Parametric model fits on only unisensory data.** We have also fitted 9 parametric models on only the unisensory data. Under unisensory task setups, the causal inference strategy is irrelevant and free parameters $(p_{\text{same}}, \beta_V, \beta_A)$ are dropped. These unisensory parametric models still differ in their priors, sensory noise, and whether the auditory-visual recalibration parameter $\rho_A$ is a free parameter or fixed to either 1 or 4/3. Overall, across the 9 parametric models fitted on unisensory data, the models with fewest parameters contain 7 parameters (Const-SingleGaussian_1 or 4/3), and the models with most parameters contain 14 parameters (Exp-GaussianLaplace, Exp-TwoGaussians).

**Parametric model names.** We name each model using the format "sensory noise – prior – causal inference strategy". For example, the model with Exponential sensory noise, Gaussian-Laplace prior, and Model Averaging causal inference strategy is denoted "Exp-GaussianLaplace-MA" from now on. For UV and UA trials, causal inference strategy is irrelevant and may be omitted from the model's name. If we additionally remove the auditory range recalibration parameter and make it a fixed constant (either 4/3 for the visual/auditory stimulus range ratio, or 1 for no recalibration), this fixed constant is added to the end of the model's name. For example, "Exp-GaussianLaplace-MA_4/3".

**Model summary.** In Table 1, we list the combinatorial set of models fitted and visualized throughout the manuscript, for both unisensory and bisensory fits.

**Table 1. Summary of models fitted.** Numbers in parentheses convey the number of free parameters incurred by each modeling component (for sensory noise, the number of free parameters is multiplied by 2 due to separate $\theta_V$, $\theta_A$). The auditory range recalibration parameter is only systematically varied in the unisensory fits (*), and always free in bisensory fits. The semiparametric model is fitted only on unisensory data (**), while its fitted sensory noise and prior functions are then used for LiftedSemiparam models fitted to bisensory data (***). The acronyms MS, MA, and PM stand for model selection, model averaging, and probability matching causal inference strategies, respectively.

| Model | Sensory noise | Prior | Aud. recalib. (unisensory*) | Causal inf. strategy (bisensory) |
|---|---|---|---|---|
| Exp-GaussianLaplace | Exponential (3 × 2) | Gaussian-Laplace mixture (3) | 1, 4/3, free (1) | MS, MA, PM |
| Exp-SingleGaussian | Exponential (3 × 2) | Gaussian (1) | free (1) | MS, MA, PM |
| Const-GaussianLaplace | Constant (1 × 2) | Gaussian-Laplace mixture (3) | free (1) | MS, MA, PM |
| Exp-TwoGaussians | Exponential (3 × 2) | Two-Gaussian mixture (3) | free (1) | MS, MA, PM |
| Const-SingleGaussian | Constant (1 × 2) | Gaussian (1) | 1, 4/3, free (1) | MS, MA, PM |
| Semiparametric** LiftedSemiparam*** | Flexible (12 × 2) | Flexible (11) | free (1) | MS, MA, PM |

## Numerical integration

The heteroskedastic sensory noise functions and non-Gaussian prior distributions used throughout the models prevents us from deriving an analytical expression for the likelihood of the human data under a certain set of free parameter values. To overcome this issue, we compute all definite integrals numerically using the trapezoidal rule, over a numerical grid defined over the variable(s) of integration.

## Model fitting

All our models have their free parameters $\theta$ fitted via maximum-likelihood estimation (MLE), which maximizes the probability of observing the participants' data [10,107]. This process is implemented as a minimization of the summed negative log likelihood (NLL) across trials, separately for each participant's dataset:

$$\text{NLL}(\theta, \text{model}) = -\log\left(p(\text{data}|\theta, \text{model})\right)$$

$$= -\log\left(\prod_{i=1}^{N_{\text{trials}}} p(r^{(i)}|s_V^{(i)}, s_A^{(i)}, \theta, \text{model})\right)$$

$$= -\sum_{i=1}^{N_{\text{trials}}} \log\left(p(r^{(i)}|s_V^{(i)}, s_A^{(i)}, \theta, \text{model})\right)$$

$$(20)$$

where $i$ denotes the trial number and $r^{(i)}$ is the response provided for the $i$-th trial (either a stimulus location estimate $r_V$, $r_A$ for localization trials, or a same/different source judgment $\hat{C}$ for BC trials).

For minimization, we used two distinct optimization algorithms, both able to handle complex non-convex landscapes with potentially multiple local optima. For optimization problems with up to 20 free parameters, we adopted Bayesian Adaptive Direct Search (BADS) [108], which is particularly effective in moderate dimension for rough loss landscapes that are relatively computationally intensive to explore. Since, like many other Bayesian optimization methods [109], BADS struggles in high dimension, we switched to CMA-ES (Covariance Matrix Adaptation Evolution Strategy) [110] for the semiparametric fits (40 free parameters). CMA-ES is generally less efficient, requiring way more evaluations than BADS to converge to a solution [108], but is still a powerful optimization method that can handle higher-dimensional problems.

PLOS Computational Biology

For each dataset (15 participants) and model, we repeated the optimization procedure multiple times (runs) from distinct starting points (initializations), to ensure a wide exploration of the loss landscape. The maximum-likelihood solution corresponds to the global minimum across all runs. The initializations were chosen as follows.

- **Semiparametric models (40 parameters).** The high-dimensional loss landscape of these models is particularly prone to local minima, requiring many optimization runs.

For each participant, we created 81 random initializations for the semiparametric model parameters, such that the three functions $\sigma_V(s), \sigma_A(s), p(s)$ are all flat across the stimulus range at various fixed values. The initial values of other model parameters are randomized uniformly within reasonable ranges. Thus, we performed $15 \times 81 = 1215$ optimization runs.

- **Lifted-semiparametric fits (9 parameters).** We performed 100 fits with different random initializations for each participant. This amounted to $15 \times 100 = 1500$ fits for each of the 9 lifted-semiparametric models.

- **Parametric models (7–17 parameters for all data fits).** A total of 21 parametric models have been fitted, corresponding to different priors $p(s)$, sensory noise functions $\sigma(s)$, causal inference strategies, being fitted on either only the unisensory data or all data, and whether the auditory range recalibration parameter $\rho_A$ is fitted, fixed at 1, or fixed at $\frac{4}{3}$. Due to longer times associated with fitting this many parametric models, we performed 10 fits with different random initializations for each participant. This amounted to $15 \times 10 = 150$ fits for each parametric model.

## Model visualization

**UV and UA tasks.** Starting from Fig 2A, we have visualized the response distributions $p(r|s)$ for UV and UA trials constructed from participants' responses (error bars) and predictive distributions from the fitted model (ribbons), the mean responses, as well as the SD of responses.

The four rows of each UV + UA figure correspond to the three visual reliability levels for UV, followed by UA. The three columns correspond to response distributions, mean response, and SD of response across human participants. We describe below the visualization process for one row.

First, we stratify responses into 7 stimulus bins according to the true stimulus location $s$, and plot response distributions separately for each stimulus bin (visualized in different colors). For the UA task, the 7 stimulus bins correspond to the 7 possible auditory stimulus locations $s_A \in \{0° \pm 5°, \pm 10°, \pm 15°\}$; for the UV task, the 7 stimulus bins equi-partition the continuous visual stimulus range $s_V \in [-20°, 20°]$.

To visualize response distributions in the first column, within each stimulus bin, we construct a separate response distribution for each participant. The response distribution is constructed by binning the participant's responses into response bins of width $3°$ with one of the bins centered at $r = 0$, counting the number of responses in each response bin, and normalizing across all response bins. We then compute the mean ± standard error of the mean (SEM) across participant response distributions for each response bin, which gives rise to the error bars (humans) and ribbons (model predictions).

To visualize mean responses in the second column, we first compute the mean response for each participant for each stimulus bin. Then, we compute the mean±SEM of the mean response across participants for each stimulus bin, which gives rise to the error bars and ribbons. Note that the error bars and ribbons have $x$-values corresponding to the stimulus bin centers.

To visualize SD of responses in the third column, we first compute the SD of responses for each participant for each stimulus bin. Then, we compute the mean±SEM of the SDs across participants for each stimulus bin, which gives rise to the error bars and ribbons.

**BC task.** Starting from Fig 6A, we have visualized the human BC data (error bars) as well as the predictions of our fitted model (ribbons).

We first stratified all trials into either center or peripheral groups, based on whether the absolute value of the sum of the two stimuli locations is below or above their median value across all trials of all participants. We then stratify trials again according to the three visual reliability levels of the visual stimuli, leading to a total of six combinations displayed in different panels.

For trials within each combination, we assign each trial to one of 9 stimulus disparity ($s_A - s_V$) bins, with bin edges at the 0th, 100/9th, 200/9th, to 100th percentile of their values across all trials within that combination. Within each stimulus disparity bin, we compute the proportion of trials for which participants or the corresponding fitted model predicts a response of $r_c = 1$ ("same"), and visualize their Mean±SEM across all participants as error bars and ribbons respectively. As before, error bars and ribbons have $x$-values corresponding to the stimulus disparity bin centers.

**BV and BA tasks.** Starting from Fig 6B, we have visualized the human BV data (error bars) as well as the predictions of our fitted lifted-semiparametric model (ribbons), depicting the mean estimation bias $\hat{s}_V - s_V$ as a function of stimulus location disparity $s_A - s_V$.

We first stratified all trials into either left, center, or right groups, based on whether the sum of the two stimuli locations lies with respect to their 25th and 75th percentiles across all trials of all participants. We then stratify trials again according to the three visual reliability levels of the visual stimuli, leading to a total of nine combinations displayed in distinct panels.

For trials within each combination, we assign each trial to one of 7 stimulus disparity ($s_A - s_V$) bins, with bin edges at the 0th, 100/7th, 200/7th, to 100th percentile of their values across all trials of all participants independent of which combination they belong to. Within each stimulus disparity bin, we compute the mean bias ($\hat{s}_V - s_V$) that the participant or the corresponding fitted model features across all trials within the stimulus disparity bin, and visualize their Mean±SEM across all participants as error bars and ribbons, respectively. As before, error bars and ribbons have $x$-values corresponding to the stimulus disparity bin centers.

Starting from Fig 6B, we have visualized the human BA data (error bars) as well as the predictions of our fitted model (ribbons). The visualization process is identical to BV data, with the only difference being that the mean estimation bias is computed for auditory stimulus as ($\hat{s}_A - s_A$).

## Model comparison

To compare different fitted models in a way that balances the ability to explain the data (likelihood) and accounts for model complexity, we used the Akaike Information Criterion (AIC) and Bayesian Information Criterion (BIC) [10,111]. These criteria are standard model comparison metrics that reward models that best describe the data while penalizing model complexity measured in terms of number of free parameters [10,111]:

$$\text{AIC(model)} = 2 \cdot \text{NLL}(\theta^*, \text{model}) + 2n_{\text{params}}$$
$$\text{BIC(model)} = 2 \cdot \text{NLL}(\theta^*, \text{model}) + \log(N_{\text{trials}}) \cdot n_{\text{params}} \tag{21}$$

where $\theta^*$ are the fitted MLE parameters for the model, and $n_{\text{params}}$ is the number of free parameters in the model. Lower AIC or BIC values indicate more parsimonious models.

For a class of different models fitted on the same data, we compared models using AIC, BIC, and raw NLL values. We report the difference in these metrics with respect to a baseline level, which is typically the best-fitting model according to the metric. In the main text we report BIC scores, leaving AIC and raw log-likelihood scores to the Appendix.

## Model recovery analysis

To assess whether our chosen model comparison metrics (AIC and BIC) could identify ground-truth data generative processes, we performed model recovery analyses. We first generate simulated datasets from multiple candidate models. For each candidate model, we generated a simulated group of 15 participants, using the human-fitted parameter values

for that model. We then fitted all candidate models to each simulated participant individually, using the same model-fitting procedure as for the human data, and performed AIC/BIC model comparison at the group level, exactly as per the analyses in the main text. We found that across simulated datasets, AIC and BIC consistently favored the ground-truth generative model, thus supporting their usage throughout the manuscript.

We performed model comparison for the unisensory setting, as it already featured most model variations of interest—sensory noise functions $\sigma(s)$ (Const or Exp), priors $p(s)$ (SingleGaussian or GaussianLaplace), and auditory range recalibration (either fixed to 1 or kept as a free parameter). We focused on a subset of models that interpolate between the vanilla Const-SingleGaussian-1 model and the best-fitting Exp-GaussianLaplace model, by varying the above model components.

The model recovery procedure was as follows. For each combination of human participant and candidate model, we extracted the stimuli displayed on each trial and the fitted model parameters. Using the candidate model and its fitted parameters, we generated responses stochastically for each trial, thus arriving at a simulated dataset. For each simulated dataset, we fit all candidate models to it (fitting procedures identical to the main text) and assessed their performance via AIC/BIC (summing over participants as usual). Finally, we examined whether the ground-truth model, fitted to its own simulated dataset, featured the lowest AIC/BIC score among all models fitted to the same dataset.

Model recovery revealed that both AIC and BIC were capable of identifying the ground-truth generative model ([Fig 14]). Specifically, for each simulated group of 15 participants, the ground-truth generative model always achieved the lowest AIC/BIC (summed across simulated participants) compared to other candidate models. Considering that the results were aggregated across subjects, AIC differences were relatively small for models that differed only up to the auditory recalibration parameter (around 1 unit per participant). However, differences in sensory noise and priors were captured by AIC prominently. In contrast, BIC was much stronger in distinguishing candidate models for each simulated dataset, so we visualize BIC model comparison results in the main text.

## Parameter recovery analysis

To check whether our model fitting procedure could identify best-fitting parameter values and assess the degree of parameter identifiability in such high-dimensional models, we performed parameter recovery using the same simulated datasets

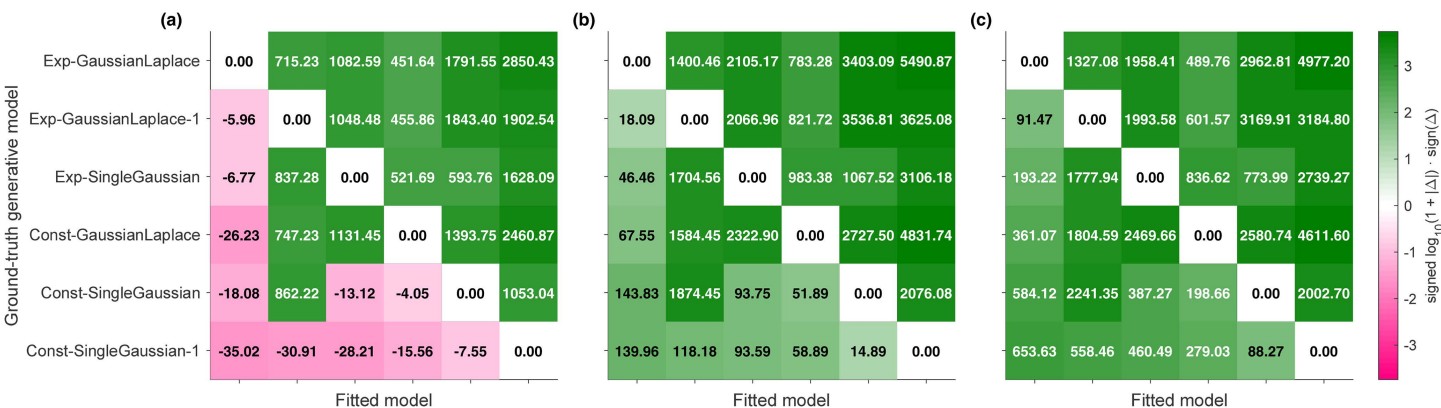

**Fig 14. Unisensory model recovery results.** Rows correspond to different simulated datasets (groups of 15 simulated participants), each named after their ground-truth generative model. Within each row, the columns are different candidate models fitted to the same simulated group (row and column orders are identical, hence omitted). Within each row, the matrix entries are NLL/AIC/BIC differences between each fitted model to the ground-truth model (the row's diagonal entry, which is always the baseline of 0). Color indicates whether the NLL/AIC/BIC difference is positive (green) or negative (pink), visualized in log scale (colorbar). Successful model recovery implies that the diagonal entries have lowest AIC/BIC within their row, and thus all matrix entries should be positive (green). **(A)** NLL differences. **(B)** AIC differences, **(C)** BIC differences.

above. We visualize the ground-truth generative parameters versus recovered parameters for the Exp-GaussianLaplace model, which was the unisensory model that i) contained the most parameters (14) and ii) was preferred by model comparison in explaining human behavior.

The parameter recovery results are visualized in Fig 15 below. Overall, we see good agreement between the ground-truth generative parameter values and their recovered values—including the sensory noise parameters ($\sigma_0, k_1, k_2$) for vision and audition, mixture prior parameters ($\sigma_s, b, w$), visual reliability scalings ($\alpha_{\text{med}}, \alpha_{\text{low}}$), the auditory recalibration parameter ($\rho_A$), and even the lapse rate ($\lambda$). Since our simulated datasets use parameters close to those found in the experiment, these results suggest that the human-fitted parameters for each model are reflective of the model's best possible performance, and that their interpretation is relatively free from parameter identification problems.

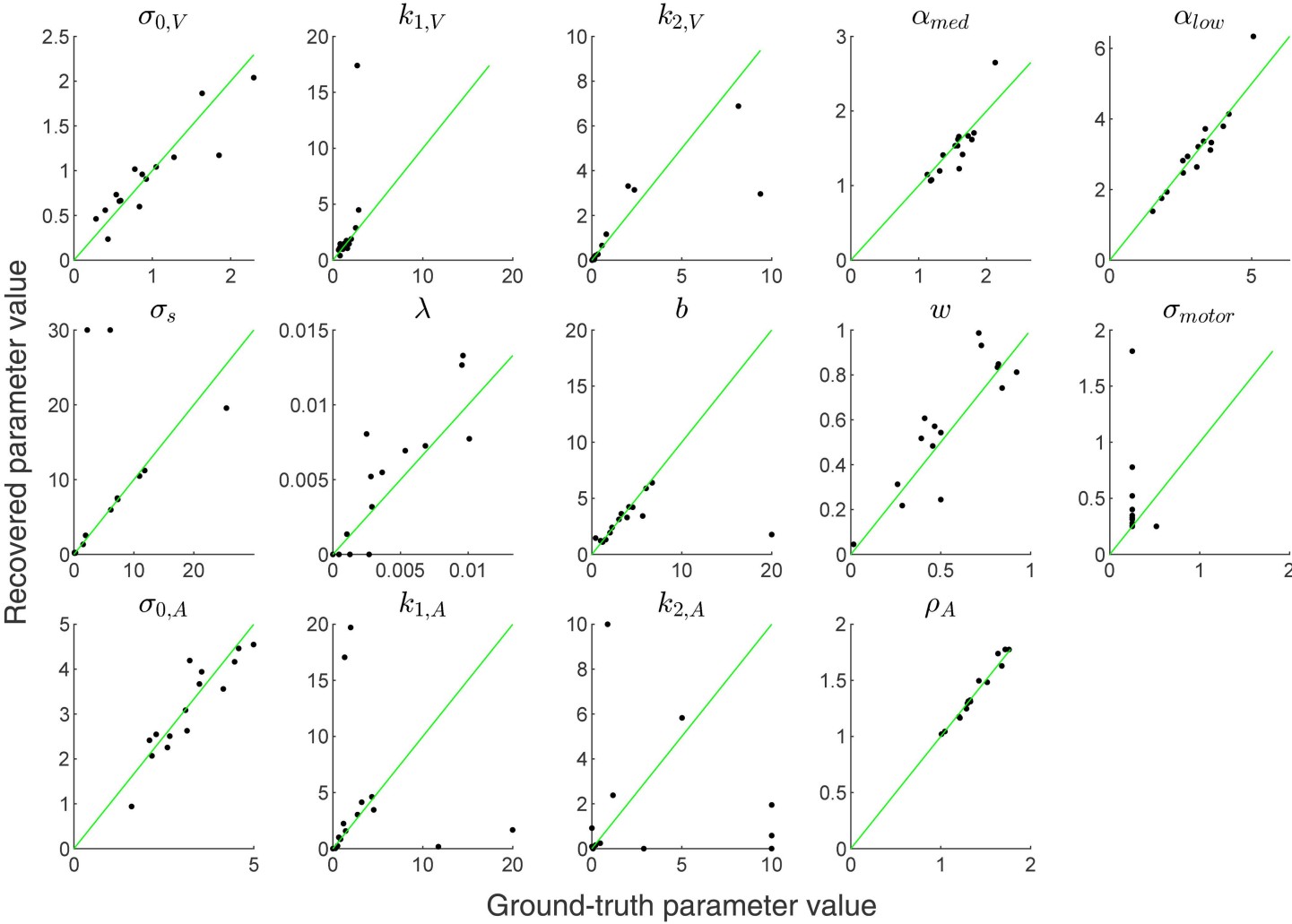

**Fig 15. Unisensory parameter recovery results for the Exp-GaussianLaplace model.** Panels correspond to different free parameters in the model (a total of 14). Data points denote the 15 simulated participants. In each panel, the horizontal axis denotes the ground-truth generative parameter values for simulated participants. The vertical axis denotes the recovered parameter values. The green solid line denotes unity (perfect parameter recovery).

## Supporting information

**S1 Appendix. Lapse distribution selection, model comparison result tables, model response distributions, and individual participant model responses.**
(PDF)

## Acknowledgments

We are grateful to Emmanouil Froudarakis for a precursor to the analysis in this paper. We thank the staff of NYU's High Performance Computing services. LA also acknowledges the research environment provided by ELLIS Institute Finland.

## Author contributions

**Conceptualization:** Trevor Holland, Wei Ji Ma, Luigi Acerbi.

**Data curation:** Trevor Holland.

**Formal analysis:** Shuze Liu.

**Funding acquisition:** Wei Ji Ma.

**Investigation:** Shuze Liu, Trevor Holland, Wei Ji Ma, Luigi Acerbi.

**Methodology:** Trevor Holland, Wei Ji Ma, Luigi Acerbi.

**Project administration:** Wei Ji Ma, Luigi Acerbi.

**Resources:** Wei Ji Ma.

**Software:** Shuze Liu.

**Supervision:** Wei Ji Ma, Luigi Acerbi.

**Validation:** Shuze Liu.

**Visualization:** Shuze Liu.

**Writing – original draft:** Shuze Liu.

**Writing – review & editing:** Shuze Liu, Wei Ji Ma, Luigi Acerbi.

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
