## [Decision Letter · Decision Letter 0]

24 Aug 2025

PCOMPBIOL-D-25-01170

Distilling noise characteristics and prior expectations in multisensory causal inference

PLOS Computational Biology

Dear Dr. Liu,

Thank you for submitting your manuscript to PLOS Computational Biology. After careful consideration, we feel that it has merit but does not fully meet PLOS Computational Biology's publication criteria as it currently stands. Therefore, we invite you to submit a revised version of the manuscript that addresses the points raised during the review process.

Please submit your revised manuscript within 60 days Oct 24 2025 11:59PM. If you will need more time than this to complete your revisions, please reply to this message or contact the journal office at ploscompbiol@plos.org. Please include the following items when submitting your revised manuscript:

We look forward to receiving your revised manuscript.

Kind regards,

Jian Liu

Academic Editor

PLOS Computational Biology

Marieke van Vugt

Section Editor

PLOS Computational Biology

**Additional Editor Comments:**

This is a promising contribution to the field of multisensory causal inference. The manuscript would be significantly improved by providing further details on the model's implementation, strengthening the experimental validation, and more fully situating the work within the broader context of previous research.

**Journal Requirements:**

Potential Copyright Issues:

i) Figures 1, and 3. Please confirm whether you drew the images / clip-art within the figure panels by hand. If you did not draw the images, please provide (a) a link to the source of the images or icons and their license / terms of use; or (b) written permission from the copyright holder to publish the images or icons under our CC BY 4.0 license. Alternatively, you may replace the images with open source alternatives. See these open source resources you may use to replace images / clip-art:

1) Please clarify all sources of financial support for your study. List the grants, grant numbers, and organizations that funded your study, including funding received from your institution. Please note that suppliers of material support, including research materials, should be recognized in the Acknowledgements section rather than in the Financial Disclosure

2) State the initials, alongside each funding source, of each author to receive each grant. For example: "This work was supported by the National Institutes of Health (####### to AM; ###### to CJ) and the National Science Foundation (###### to AM)."

3) State what role the funders took in the study. If the funders had no role in your study, please state: "The funders had no role in study design, data collection and analysis, decision to publish, or preparation of the manuscript."

4) If any authors received a salary from any of your funders, please state which authors and which funders..

6) Your current Financial Disclosure states, "The author(s) received no specific funding for this work.".

However, your funding information on the submission form indicates receiving funds.

Please indicate by return email the full and correct funding information for your study and confirm the order in which funding contributions should appear. Please be sure to indicate whether the funders played any role in the study design, data collection and analysis, decision to publish, or preparation of the manuscript.

7) Kindly revise your competing statement to align with the journal's style guidelines: 'The authors declare that there are no competing interests.'

**Reviewers' comments:**

Reviewer's Responses to Questions

**Comments to the Authors:**

Reviewer #1: This study tests key assumptions commonly made in modeling multisensory integration within the Bayesian Causal Inference framework—namely, the assumptions of noise homoskedasticity and Gaussian priors. Drawing on a novel psychophysical experiment, the authors conclude that these assumptions are not supported by empirical data. They employ a semiparametric approach to infer the shape of the sensory noise and priors, leading to a model that fits the data more accurately than conventional approaches.

This is an interesting and well-executed paper. Both the theoretical framing and experimental design are solid, and the methodological innovations are compelling. The manuscript is clearly written, and the data analyses are rigorous and thorough.

The newly proposed model indeed achieves a better fit, but at the cost of introducing more free parameters. While the authors rightly justify this with model comparisons (e.g., via BIC), I would encourage a more explicit discussion of the trade-offs involved. Specifically, models with many free parameters, even if statistically favored, may sacrifice parsimony and interpretability. This raises important questions about falsifiability: with sufficient flexibility, nearly any dataset could be accommodated. This is a general challenge in the field, not a flaw of this particular model, but it would be valuable to foreground this issue for the benefit of readers.

The “distilled” parametric models are introduced as more tractable alternatives to the semiparametric approach. However, it would help to clarify how these distilled models should be treated: Are they intended as new normative models of perception, or simply as fitting tools? If they are meant to be candidate perceptual models in their own right, more discussion is warranted on whether their specific functional forms (e.g., exponential noise, Gaussian-Laplace priors) have perceptual or neurophysiological plausibility. Otherwise, the reader is left uncertain whether to interpret these models as descriptive conveniences or theoretical proposals.

The findings offer convincing evidence for re-examining our modeling assumptions. That said, questions about generalizability remain. Would the new model outperform standard models on other datasets using similar paradigms? For example, two relevant studies with publicly available data are:

Noel, J. P., Shivkumar, S., Dokka, K., Haefner, R. M., & Angelaki, D. E. (2022). Aberrant causal inference and presence of a compensatory mechanism in autism spectrum disorder. Elife.

Mohl, J. T., Pearson, J. M., & Groh, J. M. (2020). Monkeys and humans implement causal inference to simultaneously localize auditory and visual stimuli. Journal of Neurophysiology.

Both of these studies use similar tasks but differ in key methodological aspects—particularly in how responses are collected. The current study relies on a visual pointer to report stimulus location. This may introduce asymmetries: in auditory-only trials, participants respond using visual feedback, whereas in visual and audiovisual trials, both stimulus and response are visually mediated. This difference could potentially bias modality weighting. Neither of the above studies face this issue, and showing that the new model generalizes to those datasets would significantly strengthen the present claims.

While the current model is evaluated using standard, low-level stimuli such as Gaussian blobs, it would be helpful for the authors to comment on how their modeling assumptions might adapt when dealing with more complex, naturalistic stimuli. In particular, real-world signals may not conform to Gaussian statistics, and it would be useful to discuss how the structure of priors and noise models might need to evolve under such conditions.

Finally, a recent study (Parise, 2025, eLife) introduces an algorithmic, stimulus-computable implementation of the Bayesian Causal Inference framework. This model operates with a small number of free parameters and does not rely on priors. It would be valuable for the authors to discuss how their approach compares to this algorithmic model—particularly in terms of theoretical commitments, flexibility, and the trade-off between parameter-rich fitting and process-level explanation.

Parise, C.V. (2025). A Stimulus-Computable Model for Audiovisual Perception and Spatial Orienting in Mammals. Elife.

Reviewer #2: In this manuscript by Shuze Liu et al, the authors proposed and tested a set of Bayesian observer models for visual-auditory causal inference tasks. The key idea is to development more general models that relax the assumptions of homoskedastic sensory noise and Gaussian priors. The authors proposed several variations of models, and used these models to study several unisensory and bisensory tasks. Notably, the model parameters inferred in the unisensory tasks were used to inform the model for bisensory tasks. The authors also distilled a new parametric model inspired by the results on the more complex semi-parametric models.

Overall, this is an impressive and well executed study. I found the overall results to be interesting and worthwhile. The authors proposed and empirically tested several versions of Bayesian models using behavioral data from multiple experimental tasks. The analyses and model comparisons are systematic and thorough.

I have a few modest to substantial concerns. I hope these could be addressed in a revision.

######Major points:

1. My first concern is about the overall framing of the study. The main innovation of this paper is to include more flexible sensory noise and priors to the model for modeling the visual-auditory causal inference tasks. While classic Bayesian models often make simplified assumptions of noise and priors, models with more flexible sensory noise and priors have also been proposed and tested over the past decade (e.g., Wei & Stocker, Nature Neuroscience, 2015; Polania, Woodford & Ruff, Nature Neuroscience, 2019; Prat-Carrabin & Woodford, Nature Human Behaviour, 2022; Hahn & Wei, Nature Neuroscience, 2024). In particular, Hahn & Wei (2024) described a flexible Bayesian model fitting framework that takes into account arbitrary smooth priors and noise variability. These models focused on the study of unisensory task, in particular one-d estimation tasks.

I don’t feel a direct comparison to the modeling procedures in Hahn & Wei (2024) for the unisensory cases is needed, but the paper would be stronger if the authors could engage and discuss that literature from the beginning of this paper. This would help clarify the main contributions of this paper.

Having said that, I would like to emphasize that the main models proposed for the bisensory tasks in this paper are certainty novel, and go beyond what was done previously. In particular, for these tasks, additional assumptions on the causal inference strategies need to be assumed. They authors tested three strategies, model averaging, model selection and probability matching.

2. In the semi-parametric model and the Exp-GaussianLaplace parametric model, what accounts for the near zero biases for the auditory task? It was not clear to me whether they are accounted for by the more flexible prior and noise, or by range recalibration discussed in earlier part of the paper.

One more point about the range recalibration:

This looks a bit post-hoc. I wonder if the authors have any other additional evidence for it beyond that including this mechanism can better explain the biases.

3. For the high visual reliability condition, the fits are consistent off for the [-8.6,-2.9] deg and [2.9, 8.6 ] deg conditions, in particular around the peak. Can the authors comment on this?

4. Some of the materials in Results may be merged into Methods to make the text flow better.

#####

Minor points:

Line 377-379, the authors wrote “the fits still present the same general issues highlighted previously ”. It may be useful to briefly mentioned the issues again here.

In Fig. 9, the LiftedSemiparam-PM model contains two set of numbers. This is slightly confusing. It might be better to split that into two separate rows.

The authors reported results based on a number of models and experimental conditions. While this is impressive, it also makes the paper more difficult to follow. I wonder if it would be useful to have a table to summarize these.

Reviewer #3: Liu and colleagues report an approach to estimating location-dependent sensory noise distributions and individual priors of unrestricted form. They estimate and then approximate these distributions in the context of Bayesian causal inference models fit to visual, auditory, and visual-auditory localization responses. The unusually skewed localization responses are best described by causal inference models with strongly heteroscedastic noise distributions and priors with the majority of mass in the center of the stimulus space.

The approach is interesting and relevant beyond multisensory causal inference. However, I have concerns about a few untested model assumptions, the recoverability of the two approximated distributions, and am not convinced that the rather strong claims throughout the manuscript are warranted.

The unisensory visual localization data are fascinating, very consistently skewed towards the center across different levels of visual reliability. Visual and auditory localization response variability clearly varies with distance from the center. Both characteristics evidently drive the recovered noise functions and priors. However, I could not find any publication with indications of similarly extreme heteroscedasticity (and checks of our own data routinely reveal homoscedasticity in the auditory domain) or central tendencies in uni- or multisensory visual or auditory localization. Thus, these data might well be specific to the stimuli or testing protocol. This implies that 1) the conclusion that all previous models are wrong by assuming homoscedasticity or Gaussian priors seems not warranted, and 2) it would be good to account for the possibility that the effects are to some degree driven by response rather than sensory processes.

Lapses are modeled using a uniform distribution, which makes sense for accidental mouse clicks if the cursor initially appears at random locations. However, a remark in the discussion suggests this was not the case, and that accidental clicks would tend to produce central responses. Additionally, since the stimuli are very brief, participants might miss them—perhaps because they blinked. For such lapses, smart participants would select locations in the center of the screen. Therefore, everything points towards a non-uniform lapse distribution with a sharp peak at the center.

Response noise is assumed to be independent of location. However, if participants really maintain fixation, the perception of the visual cursor should be noisier in peripheral stimulus regions. (If they moved their eyes, there’s no reason to assume visual heteroscedasticity.) Moreover, if the cursor always appeared in the center, perceptual learning might quickly create an asymmetry in sensory precision across the visual field. Therefore, it would be good to test model variants with location-dependent response noise.

The authors’ models are compared to “vanilla” models of Bayesian unisensory perception and Bayesian cross-modal causal inference. There is clearly value in fitting the simplest possible model to demonstrate its limitations in capturing the data’s characteristics. However, the comparison to a vanilla model does not clarify whether the heteroscedasticity of sensory noise or the peaky prior causes the qualitative change in predictions. The AIC values indicate that the former is the case; however, it would be helpful to illustrate (and discuss) the role of each factor in replicating the observed data. Furthermore, it should be clarified that the vanilla models are straw men and not representative of the field.

The tested models assume the existence of supramodal priors. However, this assumption is not supported by unimodal data and does not match the statistics of the natural world: objects we see clearly tend to be in the center of our vision because of poor resolution in the periphery, while many sounds we hear come from outside our visual field. The experimental stimulus distribution is supramodal (except for a difference in range) but very different from the recovered priors.

I could not find any mention of model or parameter recovery. Both checks are relevant given the high dimensionality of the models. Here, it seems especially important to show that potential trade-offs between the recovered prior, likelihood/noise distribution, and lapse and response noise distributions are under control.

Introduction and discussion of the multisensory (causal inference) literature seem to rely on a subset of earlier publications. For example, the authors state that previous studies either assumed constant sensory noise across unisensory and multisensory conditions or did not use unisensory conditions at all. On the contrary, many modeling and non-modeling studies assume differences in noise between unisensory and multisensory conditions. Similarly, the authors discuss the uniqueness of their implementation of auditory remapping, but several cited and uncited studies have implemented remapping, i.e., miscalibrated sensory signals in their models.

Previous studies that modeled both multisensory tasks, bisensory localization and bisensory causal judgment, report that while localization responses are well described by non-vanilla causal inference models, direct causal inference judgments seem to rely on heuristic decision rules (Badde et al., 2019; Hong et al., 2021, 2022, 2025). This might explain why the study did not clearly select variants that implement model averaging, as most studies that favor model averaging only fit localization data.

It is quite surprising to see that the locations of auditory stimuli influence the localization of high-reliability visual stimuli. Are participants’ common cause priors very high? Had participants been told that visual and auditory stimuli would be presented in the same location half of the time?

Including a table with each participant’s best-fitting parameters in the relevant model(s) (in the appendix) would be helpful.

Group-level averages can be misleading, despite the appeal of the curves. Some adjustments might help reassure the weary reader. Figures presenting summary statistics (e.g., bias or location-dependent SD) should show mean values and single participant statistics instead of standard errors of the mean.

If the noise distribution depends on location, is it reasonable to assume that auditory likelihoods—which cannot be inferred from population codes like in vision—are accurate?

I am struggling to reconcile the recovered priors with the existing literature. Previous studies, especially those examining the perception of durations, demonstrate that participants quickly learn simple contextual priors, such as the uniform distributions used in this experiment. Why would participants in this study rely on a prior that does not match either the distribution of the stimuli or the typical visual and auditory locations encountered in real life? Why wouldn't they learn the contextual prior over five sessions?

Reviewer comments about the style of a manuscript are subjective by nature, so whether or not the following comments are addressed is, in my view, the authors’ creative freedom.

To me, it seems unlikely that readers are able or willing to learn all of the acronyms the authors introduce and reliably retrieve them 20 pages later. I already failed at recalling them after a single paragraph. In my view, there is no good reason to force readers to learn new labels.

The manuscript is quite lengthy and sometimes reads more like a report. The authors might consider making the narrative more concise for the reader and relocating some material to the appendix.

The manuscript makes several strong claims, which set a certain tone. I personally believe that dismissing others’ work and ideas is neither necessary to demonstrate the quality of one’s own work nor helpful in making science more welcoming.

**Have the authors made all data and (if applicable) computational code underlying the findings in their manuscript fully available?**

Reviewer #1: Yes

Reviewer #2: Yes

Reviewer #3: None

PLOS authors have the option to publish the peer review history of their article (what does this mean?). If published, this will include your full peer review and any attached files.

Reviewer #1: No

Reviewer #2: No

Reviewer #3: No

**Figure resubmission:**
---

## [Decision Letter · Decision Letter 1]

20 Apr 2026

Dear Mr. Liu,

We are pleased to inform you that your manuscript 'Distilling noise characteristics and prior expectations in multisensory causal inference' has been provisionally accepted for publication in PLOS Computational Biology.

Best regards,

Jian Liu

Academic Editor

PLOS Computational Biology

Marieke van Vugt

Section Editor

PLOS Computational Biology

Please consider the reviewers' comments along with the additional suggestions below when preparing the final version. These include minor revisions and reference checks. For example, some bioRxiv preprints cited in the manuscript could be updated with their published counterparts.

Reviewer's Responses to Questions

**Comments to the Authors:**

Reviewer #1: Overall, the authors have done a good job addressing most of my previous comments, and the manuscript has clearly improved. However, one important concern remains unresolved. The manuscript uses the unimodal data to test assumptions of the BCI framework, yet the unimodal auditory data contain an unexpected finding, raising the possibility that some of the reported deviations may be specific to this paradigm.”

If the authors wish to challenge standard assumptions of the BCI model, this needs to be demonstrated on the broadest possible datasets. At present, it remains unclear whether the reported deviations reflect a general problem for the model or instead the particular experimental choices of this study. Given that relevant public datasets are available and do not suffer from the same limitations, the most convincing way forward would be to test the generalizability of the conclusions on such additional datasets, see references below. Otherwise, the claims should be explicitly limited to thisvery specific paradigm and should no longer be presented as a general challenge to standard BCI assumptions.

References:

Noel, J. P., Shivkumar, S., Dokka, K., Haefner, R. M., & Angelaki, D. E. (2022). Aberrant causal inference and presence of a compensatory mechanism in autism spectrum disorder. Elife, 11, e71866.

Mohl, J. T., Pearson, J. M., & Groh, J. M. (2020). Monkeys and humans implement causal inference to simultaneously localize auditory and visual stimuli. Journal of Neurophysiology, 124(3), 715-727.

Reviewer #2: The authors did a nice job addressing my concerns in the revision. The paper represents a nice contribution. I have no more further comments.

Reviewer #3: The authors have addressed all of my comments. From my position, no further round of reviews is needed.

I have a list of suggestions, remarks, and comments the authors might find useful.

Abstract: “they typically rely on simplifying assumptions that may not reflect the true complexity of human perception” is not really needed to highlight the current research.

Abstract: “highlight the value of making Bayesian cognitive models more data-driven”, I’d say most models are data-driven. The current study highlights the value of data- rather than theory-based model assumptions.

A figure that illustrates the effect of location-dependent noise on the measurement distributions and likelihoods would be extremely useful.

Line 63 Please add Hong et al., 2025.

Line 107 I’d have thought that the key advantage of the introduced method over that from the Simoncelli lab is that this one doesn’t require manipulations of sensory uncertainty to work (unless the semiparametric procedure hinges on the visual uncertainty manipulation in a non-obvious way). Aside from the typically tested visual features, manipulations of sensory uncertainty can be hard or impossible to achieve.

Line 114-116 Several studies in the field use localization with a pointer. Please revise the sentence as it currently evokes the impression that all researchers in the multisensory field don’t know how to measure localization, which is incorrect.

Line 127 “Our second aim is to derive a model family for priors and noise that captures the complexity of (multisensory) perception, without the need to go through the complexity of the full semiparametric approach” would sound far less patronizing.

Line 164 “These findings demonstrate the complexity of human causal inference and stimulus localization and highlight the utility of data-oriented semiparametric modeling approaches for reaching realistic modeling assumptions.” Would highlight the achievements of the current paper as well as the original sentence, but without dissing a whole field.

Line 208 What was the initial cursor position in each trial? Why was there a different claim in the previous version?

Line 271 “where the probability of each value is described by the measurement distribution” (the distribution cannot really generate values)

Line 319 Weird phrasing, I’d say “consider probability distributions defined by parametric, …, probability density functions”. Also, acronyms are capitalized.

Line 329 I’d say those priors capture expectations about the locations of visual, auditory, and audiovisual events, not about the causal structure of audiovisual events.

Line 408 “Our working hypothesis is that (a) the shape of the sensory noise function is not constant across the locations and (b) the prior over locations is not Gaussian.” Same content, but without judgment of previous modeling accounts (incl. the authors’ own ones).

Line 412 The cited papers don’t necessarily claim that sensory noise increases in the periphery; several attribute performance differences to decisional and cognitive factors.

Line 416 Broken sentence.

Line 428 The shape of the prior’s PDF and of the function scaling the noise with eccentricity?

Line 754 I, on the contrary, have deep faith in the laziness of undergraduates. A model comparison to prove one of these opposing beliefs wrong would have been nice.

Line 781 “and several causal inference studies assume an increase in sensory noise under bimodal conditions (69,97, Badde et al., 2019, Cognition; …)”

Line 807 I’m confused. If the audiovisual condition was tested before or interleaved with the unisensory ones (which seems to be the case for most blocks), the recalibration should affect auditory localization in uni- and bisensory trials. Participants typically retain recalibration effects (as if the brain stores a separate model for the experimental room).

Line 857 “Our study reveals stimulus-dependent (eccentricity-dependent) sensory noise and non-Gaussian priors as critical determinants of multisensory perception.”

Line 928 “locked to the horizontal midline” evokes the impression that the cursor appeared in the center and was moved from there. If that is not the case, please describe the initial random position of the cursor in each trial.

Line 972 Is causal inference with probability matching optimal? Which cost function does it minimize?

Eq 2 The model assumes a supramodal prior over location, i.e., p(s)=p(s_A)=p(s_V), correct?

**Have the authors made all data and (if applicable) computational code underlying the findings in their manuscript fully available?**

Reviewer #1: Yes

Reviewer #2: Yes

Reviewer #3: None

PLOS authors have the option to publish the peer review history of their article (what does this mean?). If published, this will include your full peer review and any attached files.

Reviewer #1: No

Reviewer #2: No

Reviewer #3: No

---

## [Editor Report · Acceptance letter]

PCOMPBIOL-D-25-01170R1

Distilling noise characteristics and prior expectations in multisensory causal inference

Dear Dr Liu,

I am pleased to inform you that your manuscript has been formally accepted for publication in PLOS Computational Biology. Your manuscript is now with our production department and you will be notified of the publication date in due course.

With kind regards,

Anita Estes
